# Super-silencers are crucial for development and carcinogenesis in B cells

Di Huang[1], Hanna M. Petrykowska[2], Dhaneshwar Kumar[3], Lela Kardava [4], Susan Moir [4], Behdad Afzali [3], Laura Elnitski [2] ✉ & Ivan Ovcharenko [1] ✉

The strength of the repressive histone H3K27me3 signal varies across silencers. Focusing on regions with unusually strong signals—super-silencers—we show that B-cell super-silencers are initially linked to gene upregulation in development, with target genes highly expressed in stem cells. About 13% of B-cell super-silencers convert to super-enhancers in B-cell lymphoma; 22% of these recur in over half of patients. Genes like BCL6 and BACH2 tied to these conversions are downregulated faster by JQ1, a super-enhancer-disrupting anticancer agent. Super-silencers are enriched for B-cell cancer-associated variants—both somatic and germline—and translocation breakpoints, exceeding levels in other regulatory elements like CTCF binding sites. Over 80% of B-cell lymphoma t(3;14)(q27;q32) translocations fuse BCL6 super-silencers with enhancer-rich regions. Super-silencer repression depends on CpG content: CpG-rich elements block promoter–enhancer contacts; CpG-poor − inhibit looping. These findings highlight super-silencers' key role in B-cell regulation and suggest their alteration may be a primary factor of B-cell carcinogenesis.

DNA regulatory elements (REs) play a crucial role in gene transcription by recruiting diverse transcription factors (TFs) to either initiate, prepare, or inhibit gene transcription[1–3]. Silencers, a type of negative REs, accomplish this by establishing repressive chromatin marks around their target promoters[4,5] or by competing with these promoters for TF binding[6–8]. Predominantly found in metazoan species, such as yeast[9], *Drosophila*[10], mammals (including humans)[11], silencers are vital for defining essential phenotypic traits[12–14]. Recent developments in massively parallel reporter assays (MPRAs) and computational analysis tools have facilitated large-scale detection of silencers in human cells and highlighted their pivotal roles in gene regulation[15–17].

Evidence has increasingly linked silencers to complex diseases[18]. For example, a mutation in the minor allele at the single-nucleotide polymorphism (SNP) rs851982 deactivates a silencer in breast cells, triggering overexpression of *ESR1* and *RMND1* in breast cancer cells[19]. Similarly, a mutation in the breast cancer risk SNP rs75915156 upregulates the oncogene *CCND1* in breast cancer cells[20].

The concept of "super-enhancers" (SEs), composed of several interacting enhancers and featuring an unusual abundance of coactivators and mediators, was introduced to address the challenge of annotating and prioritizing enhancers across various cell types[21]. SEs, characterized by strong histone H3 lysine 27 acetylation (H3K27ac) signals, exhibit significant association with cell-specific processes[22] and enrichment in the master TF bindings[23]. The high density of disease-associated variants and the common acquisition of SEs near oncogenes during tumorigenesis underline the crucial role of SEs in controlling core cellular functions and cell identity.

Conversely, genomic regions with significant histone H3 lysine 27 trimethylation (H3K27me3) signals, termed here as super-silencers (SSs)[24,25], are on the opposite end of the regulatory spectrum. Previously identified in the chronic myelogenous leukemia cell line K562[25], SSs, which were named as H3K27me3-rich regions (MRRs) hitherto, interact frequently and significantly suppress their target genes. Knockout of SSs results in local topologically associated domains

[1]Intramural Research Program, National Library of Medicine, National Institutes of Health, Bethesda, MD, USA. [2]Translational and Functional Genomics Branch, National Human Genome Research Institute, National Institutes of Health, Bethesda, MD, USA. [3]Immunoregulation Section, Kidney Diseases Branch, National Institute of Diabetes and Digestive and Kidney Diseases, National Institutes of Health, Bethesda, MD, USA. [4]Laboratory of Immunoregulation, National Institute of Allergy and Infectious Diseases, National Institutes of Health, Bethesda, MD, USA. ✉e-mail: elnitski@mail.nih.gov; ovcharen@nih.gov

(TADs) disruptions and alternate chromatin loops, inhibiting tumor cell growth[24,25]. These findings emphasize the importance of SSs in maintaining cell identity in K562 cells.

However, key biological characteristics of SSs, such as their distinct function and disease relevance compared to other silencers, the molecular mechanisms and their potential contribution to chromatin folding and regulatory domains establishment and maintenance, remain unclear. Here, our detailed investigation of GM12878, a lymphoblastoid cell line, using transcriptomic, chromatin accessibility, GWAS and evolutionary conservation data, illuminates the functional significance and silencing mechanisms of SSs, suggesting a link between the loss of these mechanisms and B-cell lymphoma.

## Results
### Distinguishing super-silencers

We previously designed computational methods to identify silencers using DNA sequences and reported that silencers, instead of being randomly distributed, selectively cluster in the loci of tissue-specific genes[3]. Similar to H3K27ac signals, H3K27me3 signals feature an asymmetric distribution with a long-right tail (Fig. 1a), indicating that a small subset of silencers exhibits a significant intensity of H3K27me3 modifications. We applied the Rank Ordering of Super-Enhancers (ROSE), a classical super-enhancer-identification method[23], to the H3K27me3 signals on 34,605 silencers (averaging 1492 bp) detected using our deep learning model in GM12878 cells (see "Methods"). Consequently, we detected 879 SS regions, which collectively host 4,617 constituent silencers (Fig. 1b and Supplementary Fig. 1, and Supplementary Data 1). Notably, nearly 66% of constituent silencers in our SS regions overlapped with H3K27me3-rich regions (MRRs) reported in the previous study[25], representing a significant enrichment compared to the overlap observed in typical silencers (35%) and enhancers (9%) in GM12878 cells (binomial test $P < 10^{-200}$, Supplementary Fig. 2). The discrepancy between our SSs and MRRs is primarily due to different silencer pools used for SS detection. While MRRs were determined from all peaks of a GM12878 H3K27me3 ChIP-seq analysis[25], our SSs were profiled on GM12878 silencers that were predicted using a deep learning model from H3K27me3 ChIP-seq peaks (see the "Methods"). The predicted silencers are more often associated with lowly-expressed genes than H3K27me3 ChIP-seq peaks used in the MRR study (Wilcoxon rank-sum test $P < 10^{-10}$), as are our SSs in comparison to MRRs (Wilcoxon rank-sum test $P < 10^{-10}$, Supplementary Fig. 2). For example, the loci of bottom−10%-expressed genes host 18.0% of the predicted SSs and 12.9% of MRRs (binomial test $P = 10^{-18}$). On average, each SS is 36 kb long and consists of 5.25 individual silencers. In this study, individual silencers located within SS regions are referred to as SS "components" for convenience, while the silencers not located within SS regions are named typical silencers (TSs).

In the GM12878 cell line, over 60% of SS components and TSs reside in intergenic regions, while approximately 45% of SE components and typical enhancers (TEs, i.e., the enhancers located not in super-enhancer regions) are intergenic (Fig. 1c). Also, 27% of SS components overlap CpG islands (CGIs) by more than 200 bp. This percentage, ranking second only to gene promoter regions (42%), is significantly higher than TEs, SE components and TSs (-17%; binomial test $P < 10^{-10}$, Fig. 1c and Supplementary Fig. 3). Furthermore, SS components demonstrate significantly lower methylation levels than TEs, SE components and TSs in both CGI and non-CGI cases (Student's $t$-test $P < 10^{-10}$, Fig. 1d and Supplementary Fig. 4). Low DNA methylation levels in silencers (especially SS components) reflect the mutual antagonism between DNA methylation and H3K27me3 modification, which, at least in normal cell types[26], supports the crosstalk between these repressive epigenetic marks for transcription, which is essential for precisely tuning gene transcription programs[27].

The presence of elevated evolutionary sequence conservation has long been recognized as an indicator of functional importance in noncoding sequences[28]. In the GM12878 cell line, 13% of SS component sequences overlap genomic regions conserved in placental clades[29], which is significantly higher than 7.0% of TEs, 7.7% of SE components, 8.5% of TSs, and 4.3% of randomly selected DNase-seq peaks in other cell types (binomial test $P < 10^{-10}$, Supplementary Fig. 5). Also, the occurrence of common SNPs per kb in SS components (4.62) is lower than in TEs, TSs, and background sequences ($P < 10^{-10}$), although it is comparable to 4.61 in SEs (Supplementary Fig. 5). These combined analyses indicate that the purifying selective pressure on SS components has been strong during mammalian speciation and modern humans, suggesting a pivotal biological role of SSs.

### SSs are associated with a large decrease in gene expression

We sought to evaluate the regulatory impact of SSs by analyzing genes close to SS components. We found that, on average, both SSs and TSs were linked to lower gene expression levels than enhancers. SSs are associated with the lowest levels of gene expression among all examined silencers and enhancers (Wilcoxon rank-sum test $P < 10^{-10}$, Fig. 1e). In addition, genes having chromatin contacts with SS components and TSs, as detected in Hi-C experiments[30,31], are expressed significantly lower than those linked to enhancers and all assayed genes ($P < 10^{-10}$, Fig. 1f). These results support the idea that silencers, especially SSs, have a repressive influence, as suggested before[25]. Moreover, genes proximal to and/or contacting with SS components exhibit higher expression variation across cell/tissue types, i.e., greater tissue-specificity, compared to genes associated with TSs or enhancers ($P < 10^{-5}$, Fig. 1g, h). With the associations of the lowest gene expression levels and highest tissue-specificity, SSs among silencers can be analogized to SEs among enhancers. Since SEs have been widely reported to play crucial roles in gene regulation[22,32], SSs are worthy of further investigation.

We used the results of ATAC-STARR-seq (an assay of transposase-accessible chromatin using sequencing with a self-transcribing active regulatory region) to assess the regulatory activity of silencers. ATAC-STARR-seq measures transcriptional activities of accessible DNA segments as the ratios of output RNA reads to input DNA reads, where a significantly low negative ATAC-STARR-seq activity score indicates a "silent element"[33]. TSs and SS components have significantly lower ATAC-STARR-seq activity scores than TEs and SE components (average scores: −0.60 in TSs and −0.59 in SS components, vs. 0.23 in TEs and 0.32 in SE components, Student's $t$-test $P < 0.0001$, Fig. 1i).

Moreover, SS components have similar ATAC-STARR-seq activity scores to TSs ($P = 0.84$), which leads to the hypothesis that the cooperation among SS components clustering in the same SS region results in a significant silencing effect, in a manner similar to that of SE components collectively leading to a significant activating effect[34]. To assess cooperativity among SS components within an SS region, we examined tissue-specificity correlations between individual silencers, assuming that collaborating enhancers and silencers synchronize their activities across tissues or cell types. After combining ChIP-seq signals from H3K27ac (defining positive regulators) and H3K27me3 (defining negative regulators) at silencers across 56 human non-GM12878 cell types (see Supplementary Information), we observed that individual SS components within the same SS regions had a significantly higher correlated activity than the background regions consisting of either TSs selected to match the distances between SS components or randomly selected silencers (Wilcoxon rank-sum test $P < 10^{-100}$, Fig. 1j). Furthermore, within the same SS regions, individual SS components showed greater similarity in ChIP-seq TF binding profiles than either of these two background sets ($P < 10^{-5}$, Fig. 1k). On average, 13.5% of TFs demonstrated binding in at least two SS components within the same SS region, which is significantly higher than 11.9% of the TFs in the distribution-matching regions, and 9% of TFs among the randomly

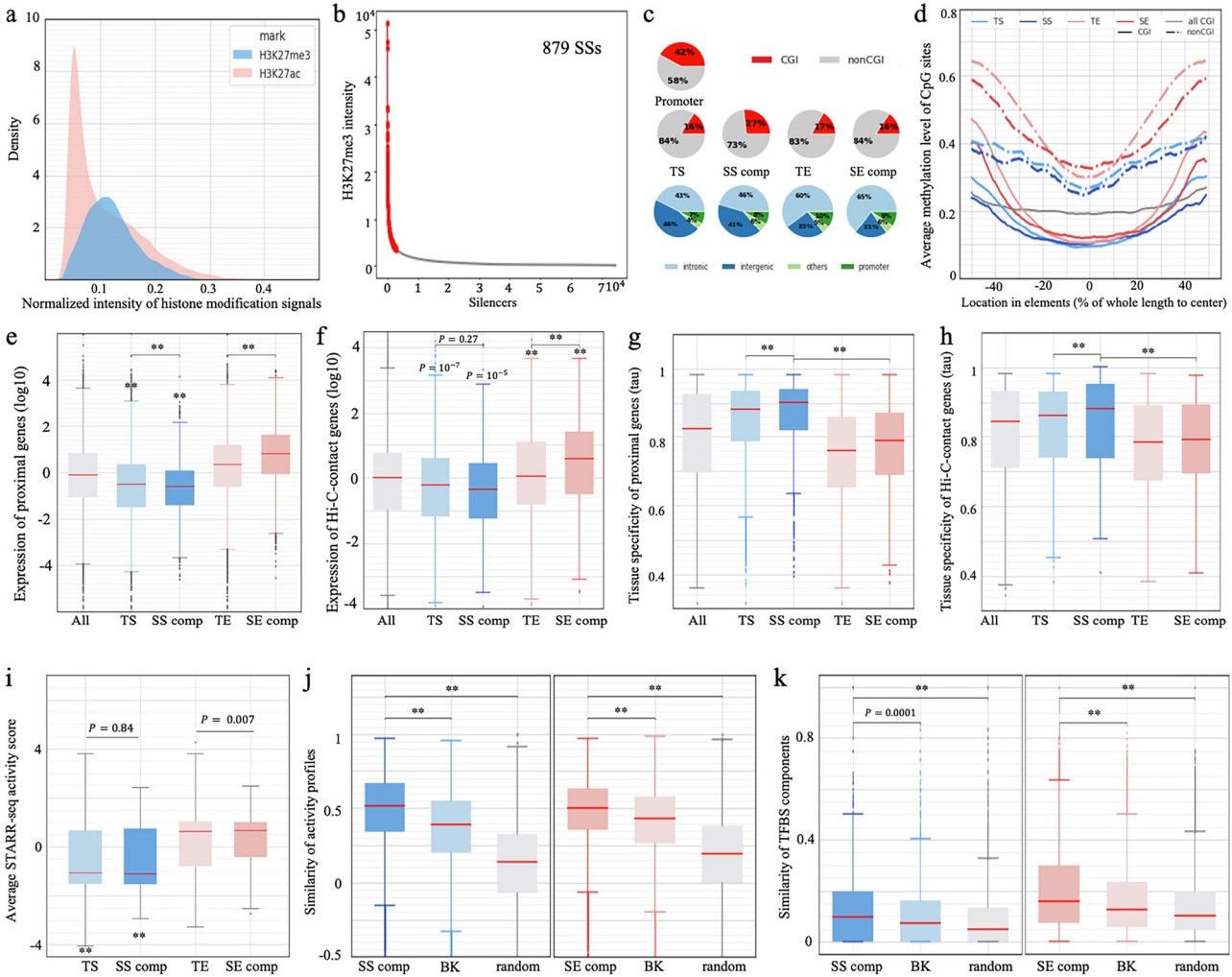

**Fig. 1 | Identification of SSs in GM12878 cells. a** Distribution of H3K27me3 ChIP-seq intensity in the silencers (blue) and of H3K27ac in the enhancers (pink). Intensity levels were linearly normalized to have a maximum of 1 for display purposes. **b** Results of ROSE with the intensities of H3K27me3 in GM12878 silencers. **c** Genomic distribution and overlap with CGIs of different enhancer and silencer types. **d** Average methylation levels of CpG sites in different enhancer and silencer types. Expression of genes (**e**) most proximal to and (**f**) having contacts with different enhancer/silencer types. Tissue-specificity of genes (as measured by *tau*) (**g**) most proximal to and (**h**) having contacts with different enhancer/silencer types. In (**e–h**), "All" represents all genes profiled in RNA-seq data. "SS comp" and "SE comp" denote SS and SE components, respectively. *P* values (**$P < 10^{-10}$) were determined against "All" (i.e., all genes assayed in RNA-seq experiments, see Supplementary Information) using two-sided Wilcoxon rank sum tests without adjustment ($n = 14,388$). The SS components (represented by dark blue) are frequently located close to or target the genes lowly expressed in GM12878 and with a high tissue specificity. **i** Average ATAC-STARR-seq scores of different enhancer and silencer types. *P* values (**$P < 10^{-10}$) were determined against "TE" using two-sided

Wilcoxon rank sum tests without adjustment ($n = 645$ for "SS comp"; 3848 for "TS"; 3726 for "SE comp" and 3848 for "TE"). **j** Activity similarity among SS components within the same SS regions across cell types. **k** TFBS similarity between SS components located within the same SS regions across cell types. In (**j, k**), "SS comp" and "SE comp" represent SS and SE components, respectively. In the left panel of (**j, k**), "BK" is the TSs randomly selected to match the distribution of distances between SS components, and "random" represents the randomly selected silencers. Similarly, "BK" and "random" in the (**j, k**) right panel are SE-distribution-matching and randomly selected TEs, respectively. In (**j, k**), **$P < 10^{-10}$ were determined against "BK" by two-sided Wilcoxon rank sum tests without adjustment. In the left panels, $n = 13,671$ for "SS comp", 15,028 for "BK", and 29,170 for "random" in left panels, while $n = 35,518$ for "SE comp", 9728 for "BK", 24,562 for "random" in right panels. In (**e–k**), the center line (in red) in a box shows the median; the box bounds represent the lower and upper quartiles; the whiskers extend to the minima and maxima points up to a maximum of 1.5× the interquartile range; and dots denote outliers.

selected silencer regions (binomial test $P < 10^{-3}$). These results, indicating strong correlation of TFBS profiles and high similarity in dynamic activity among individual SS components, suggest that these elements act in a synchronized manner.

These findings show a functional and mechanistic analogy between SSs and SEs. Also, components within the same SE region display stronger correlated activity and TFBS similarity than distribution-matching TEs and randomly selected enhancers (Fig. 1j, k). It suggests that high local cooperativity could be one of the mechanisms by which super-regulatory elements (i.e., SS and SE regions) achieve the pronounced regulatory impact.

We expanded our analysis to include three additional cell types (i.e., hESC H1, HepG2, and K562). On average, we identified about 6000 SS components per cell type. Genes proximal to or linked with these elements consistently exhibit the most significant downregulation levels among all examined genes ($P < 10^{-10}$, Supplementary Fig. 6). Furthermore, in the cell types K562 and HepG2 where the regulatory impact of genome sequences has been assessed in Sharpr MPRAs[35], the predicted SS components and TSs are significantly enriched with negative Sharpr scores, while predicted enhancers exhibit positive Sharpr scores on average ($P < 10^{-10}$, SS components and TSs vs SE components and TEs, Supplementary Fig. 6). These scores support the

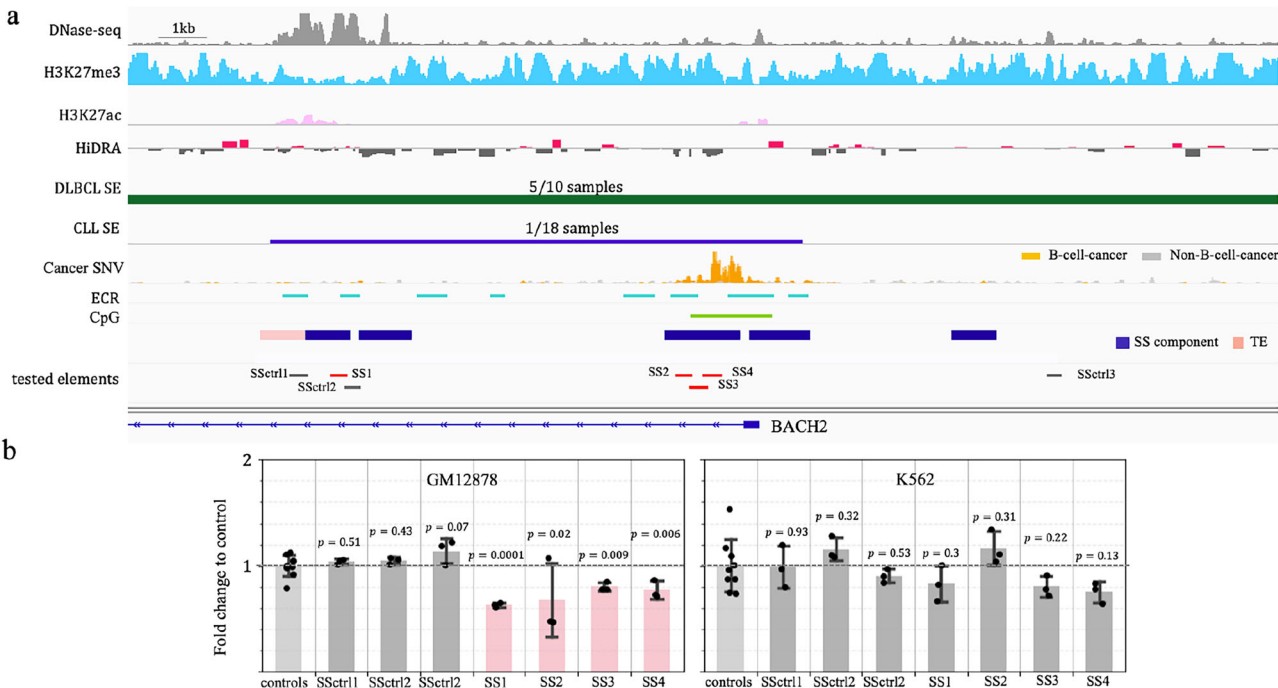

**Fig. 2 | Experimental validation of SSs. a** Elements tested in the experiments, accompanied by the genomic and epigenetic profiles of the regions hosting these elements in normal and cancer genomes. The numbers in the track of "DLBCL SE" and "CLL SE" are the numbers of patients having the corresponding SEs out of all patients. **b** Luciferase activities of the tested SSs in GM12878 and K562. Data are presented as the median ± SD ($n = 9$ for controls, and $n = 3$ for SSctrl1 ... SSctrl3 and SS1... SS4, i.e., the sequences located within the SS). All these samples are biological replicates. $P$ values were determined against control elements by two-sided $t$ tests without adjustment. Source data are provided in Supplementary Data 3.

repressive impact of silencers and activating influence of enhancers. These results, aligning with those observed in GM12878 (Fig. 1), support the negative regulatory function of predicted silencers, especially SSs, across different cell types.

We further examined the predicted SS components from additional views. Firstly, TSs and TEs show a higher density in gene deserts than their super element counterparts (Supplementary Fig. 7a). Secondly, 20.8% of SS components are proximal to cell-specific genes, significantly exceeding 14.7% expected among TSs and 11.8% among enhancers (binomial test $P < 10^{-10}$, Supplementary Fig. 7b). In contrast, SS components are notably less abundant in housekeeping gene loci (6.4% of SS components vs 9.5% of TSs vs 15% of enhancers, $P < 10^{-10}$, Supplementary Fig. 7c). Thirdly, the Jaccard indices of SS components across these cell types (i.e., the fractions of SS components shared between cell types) average 0.039, significantly lower than 0.095 of TSs, 0.084 of SEs, and 0.129 of TEs ($P ≤ 0.01$, Supplementary Fig. 8a). Additionally, the Jaccard indices of SSs between GM12878 and B cell cancer cell lines average 0.034, significantly lower than those for enhancers ($P ≤ 0.01$, Supplementary Fig. 8b). These results consistently hint the high specificity of silencers (together with their associated genes), particularly SSs, to one examined cell type or cellular context.

## Experimental validation of silencing activity of *BACH2* and *BCL6* SSs

*BACH2* is essential for the development of B-cells[36] and its overexpression has been regarded as a biomarker of diffuse large B cell lymphoma (DLBCL)[37]. There is an 18.1 kb-long SS located in the *BACH2* locus, hosting a large density of B-cell cancer mutations. *BACH2* SS components show loss of function activity, and the *BACH2* SS acts as a SE in multiple DLBCL patients[38] (Fig. 2a). To dissect the regulatory activity of *BACH2* SS components, we used luciferase reporter assays specifically designed to detect silencers[3]. We divided *BACH2* SS components into four segments, SS1...SS4, guided by the location of DNase-seq peaks and

evolutionary conservation between human and mouse genomes (see "Methods", Fig. 2a). Segments flanking the predicted silencers (SSctrl1 and SSctrl2) and another DNase-seq peak within the *BACH2* locus (SSctrl3) were selected as negative controls. All negative controls displayed H3K27me3 ChIP-seq signals comparable to TSs (Fig. 2a).

In GM12878, all four tested SS segments demonstrated a significant decrease in luciferase activity of reporter genes (Student's $t$-test, $P < 0.05$, Fig. 2b). All selected controls showed no regulatory effect on transcription of reporter genes despite flanking tested SS components having H3K27me3 modifications in GM12878 cells. Moreover, none of the examined SS segments caused a significant transcriptional change when examined in K562 cells.

Additionally, among the six examined SS segments located in the *BCL6* locus, three exhibit repressive regulatory activity in GM12878 cells ($P ≤ 0.05$ vs control elements, Supplementary Fig. 9). Combined, 70% of the ten examined SS segments in the *BACH2* or *BCL6* locus were experimentally validated as silencers in GM12878. Moreover, three out of six GM12878 SS segments function as a silencer in primary B cells (Supplementary Fig. 9a). Two of them (SS8 and SS9) have repressive influence in both GM12878 and primary B cells, hinting at a high degree of similarity in silencer profiles between these two B cell types. To further assess this similarity, we compared the silencer profiles between GM12878 and the primary B cell biosample documented in the ENCODE project (see Supplementary Information). On average, over 33% of GM12878 SSs function as SSs in primary B cells, a significantly greater overlap compared to only 6.7% of these elements acting as SSs in other cell types, including hESC H1, HepG2, and K562 ($P = 0.004$, Supplementary Fig. 9b). Similarly, TSs, SEs and TE profiles in GM12878 display substantial overlaps with their counterparts in primary B cells ($P ≤ 0.04$). Notably, GM12878 SSs and SEs, the regulatory element types critical for maintaining cell identity, show more enrichment among their counterparts in primary B cells than TSs and TEs. These high similarities underscore GM12878 as a suitable model for

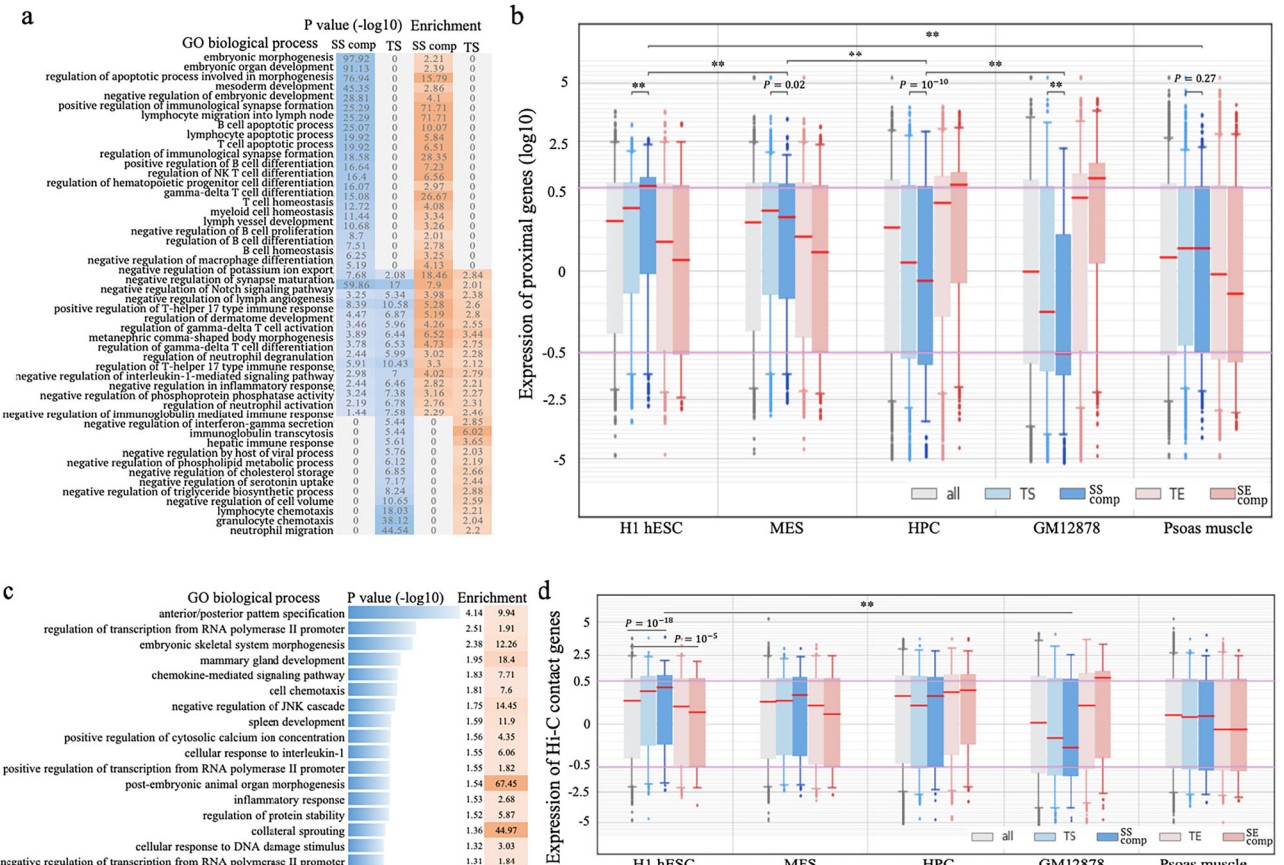

**Fig. 3 | Functional analysis on GM12878 SSs. a** Biological processes associated with SS components and TSs. These are the evaluation results of GREAT[46] in which two-sided binomial tests were performed. *P* values presented were the results without adjustment. **b** Expression of genes associated with different REs across the cell types relevant to B-cell differentiation and the Psoas muscle (used as reference). *P* values above the boxplot (**$P < 10^{-10}$) were determined against all genes assayed in RNA-seq experiments ($n = 14{,}388$, see Supplementary Information) using two-sided Wilcoxon rank sum tests without adjustment. **c** Top-ranked biological processes enriched by the genes in which the promoters interact with SS components. These are the evaluation results of DAVID[95] in which Fisher's exact tests were

used. *P* values presented were the results without adjustment. Analysis results for TSs are presented in Supplementary Fig. 11. **d** Expression of genes linked with different REs by Hi-C contacts. *P* values (**$P < 10^{-10}$) were determined using two-sided Wilcoxon rank sum tests without adjustment ($n = 14{,}388$ of all genes assayed in RNA-seq experiments, see Supplementary Information). In (**b, d**), the center line (in red) in a box shows the median; the box bounds represent the lower and upper quartiles; the whiskers extend to the minima and maxima points up to a maximum of 1.5× the interquartile range; and dots denote outliers. Throughout the figure, "SS comp" and "SE comp" represent SS and SE components, respectively.

studying gene regulation in B cells[39,40], employed as a reference point for normal B cell behavior in B-cell cancer studies[41,42].

## SSs preferentially populate the loci of developmental genes

To gain insight into the biological function of SSs, we investigated the genes flanking SSs. In GM12878, top-ranked SSs are proximal to the development-related genes, including *PKNOX2*[43], *NEUROG3*[44], and *PAX2*[45] (Fig. 1b). Functional evaluation of all SSs by the Genomic Regions Enrichment of Annotation Tool (GREAT)[46] demonstrates that, while SSs and TSs share significant associations with immune-related functions (such as negative regulation in inflammatory and immune response), SSs are uniquely associated with developmental processes. For example, SS components are 2.2-fold enriched in the loci of embryonic morphogenesis genes ($P = 10^{-97}$; GREAT), and 3.3-fold in gene loci active during B-cell differentiation ($P < 10^{-6}$), and 2.8-fold in the loci of lymph vessel development genes ($P = 10^{-6}$) (Fig. 3a and Supplementary Fig. 10). It is important to clarify that this enrichment analysis does not indicate whether B-cell SSs are active during development or if super-silencing activity is acquired by differentiated cells to selectively repress the expression of associated developmental genes.

To further delineate the developmental association of SSs, we utilized gene expression profiles. Briefly, B-cell development is characterized by a progressive gain of cell fate determination,

encompassing a series of cell lineages from embryonic stem cells (ESCs) to mesoderm to hematopoietic progenitor cells (HPCs) to common lymphoid progenitors to B-cells during embryonic development[47]. In adulthood, the development of B-cells starts from HPCs originating from the bone marrow. Limited by data availability, we approximated B-cell development steps using H1 hESC, mesendoderm (MES), HPC, and GM12878 cell lines. While GM12878 enhancers are associated with significantly high gene expression levels only in blood cell types (i.e., HPC and GM12878), GM12878 silencers are associated with high gene expression levels only in stem cells (i.e., H1 hESC and MES). Furthermore, SS components are associated with the highest average gene expression level, which was 1.5-times higher than that of TS-flanking genes and over 2.5-times higher than those of the genes next to SE components and TEs (Wilcoxon rank-sum test $P = 10^{-50}$, Fig. 3b). This overexpression trend diminishes as B-cell development proceeds beyond H1 hESC; for example, the average gene expression of SS-flanking genes descended to 0.93 times that of TS-flanking genes in MESs ($P = 0.02$), 0.76 times in HPCs ($P = 10^{-10}$), and 0.5 times in GM12878 ($P = 10^{-37}$, Fig. 3b), suggesting a gradual acquisition of super-silencing activity during development, starting from early embryonic development.

Similarly, SS-component-contact genes (as detected in Hi-C experiments[30,31]) frequently take part in developmental processes,

such as embryonic skeletal system morphogenesis and spleen development (Fig. 3c), while TS-contact genes are significantly associated with the responses to stimuli, drug, and wounding (Supplementary Fig. 11). Consistent with our previous findings, SS-component-contact genes are highly expressed in H1 hESCs but lowly expressed in GM12878 cells, showing 1.3-times higher expression in H1 hESCs ($P = 0.004$) and 0.6-times lower expression in GM12878 cells ($P = 0.07$) as compared with TS-contact genes. In addition, we also checked gene expression levels in the psoas muscle, a solid tissue barely relevant to the development of B cells. Genes associated with or targeted by GM12878 TSs and/or SS components show an insignificant expression difference from the other genes in the psoas muscle, indicating a high tissue specificity of these silencers and their target genes to GM12878 cells (Fig. 3b, d). Taken together, these results support the significant role of SSs in the development and differentiation of B cells.

It is important to note that Hi-C data reports Hi-C contacts for the promoters of less than 4000 genes, 19.6% of all examined genes. In addition, the genes with reported Hi-C contacts have higher expression levels than all genes (average expression levels of all versus Hi-C genes: −0.14 versus −0.06; Wilcoxon rank-sum test $P < 10^{-10}$, Supplementary Fig. 11), implying that lowly-expressed genes are depleted of Hi-C contacts, and thus are frequently excluded from Hi-C-based gene expression analyses. This could partially explain the insignificant expression difference between the genes having TS-contacts and having SS-contacts (Fig. 3d).

## SS components are enriched for mutations associated with a panel of diseases, especially cancers

We also utilized annotated SNPs to evaluate the functional implications of SSs. We collected genome-wide association study (GWAS) SNPs from the National Human Genome Research Institute (NHGRI)

catalog[48] and added SNPs in tight linkage disequilibrium (LD $r^2 > 0.8$) to these SNPs (see "Methods"). All GM12878 enhancers and silencers hosted a higher density of GWAS SNPs than the whole human genome (binomial test $P < 10^{-6}$), in Fig. 4a, suggesting functional importance of these elements. Especially, these elements are enriched in immune system-associated SNPs in comparison to randomly selected DNase-seq peaks in other cell types ($P < 10^{-10}$, Fig. 4b), which is consistent with the fundamental role of B cells in the immune system.

The pronounced developmental association of GM12878 SS components led us to evaluate the enrichment of the SNPs associated with B-cell cancers (i.e., lymphoma and leukemia) since cancers often arise from a malignant gain of "stemness" in normal cells[49]. Among all tested enhancer/silencer sets, SS components are most enriched in CLL (chronic lymphocytic leukemia) associated SNPs ($P < 10^{-10}$ vs the background, Fig. 4b, c), including the high-CLL-risk SNP rs13401811 and rs9308731[50]. SS components are also significantly enriched in B-cell-lymphoma-associated SNPs ($P = 10^{-22}$ versus the background). One of these SNPs, rs3806624, holds the highest association with Hodgkin's lymphoma in the host LD block, at which the mutation of the major allele A to the disease allele G weakens the binding affinity of the repressor TF p53 and increases the risk of Hodgkin's lymphoma[51,52].

SNPs with strong phenotypic influence are repeatedly identified in independent studies and are referred to as replicated SNPs[53]. Immune-system-associated SS-component SNPs (43%) were reported in multiple immune-related studies, which is significantly higher than immune-system-associated SNPs in TSs (28%) and in enhancers (35%; binomial test $P < 10^{-10}$, Fig. 4d). Remarkably, B-cell-cancer-associated SS-component SNPs (78%) are also replicated, which is twice as high as those in the other enhancer and silencer types ($P < 0.001$, Fig. 4d). These significant enrichments and high proportion of replicated SNPs

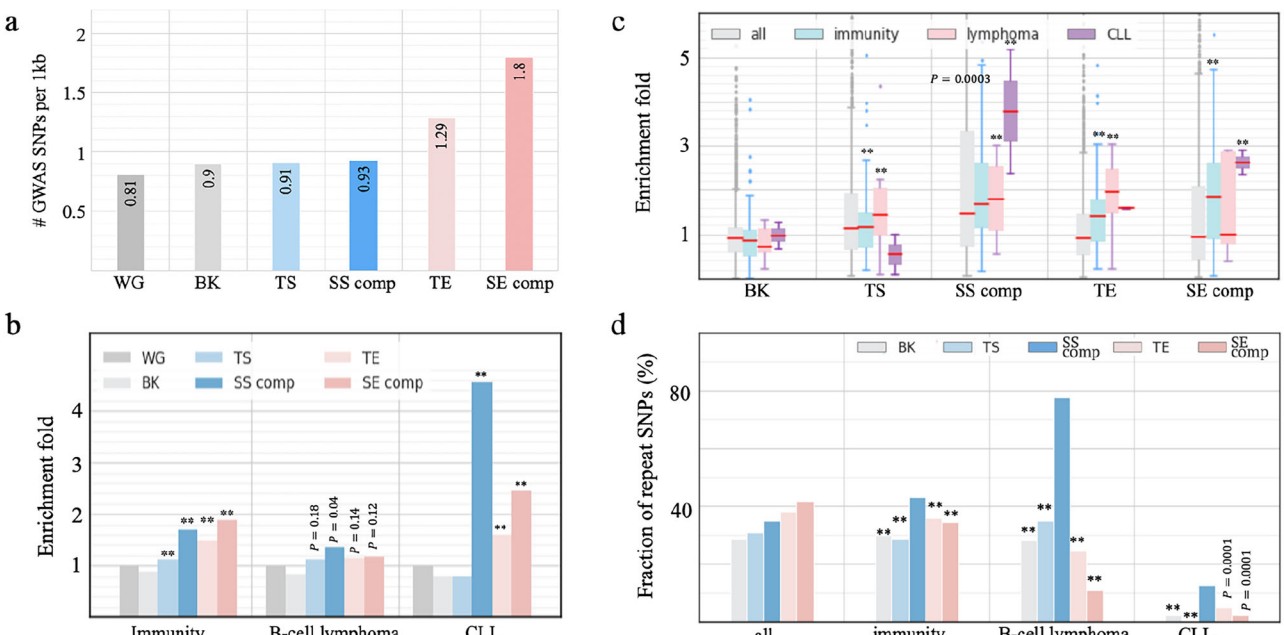

**Fig. 4 | Disease association of SSs.** Enrichment of GWAS SNPs associated with (**a**) all traits and (**b**) traits relevant to immunity, B-cell lymphoma, and CLL. "WG" is the whole human genome, and "BK" is the background sequences, i.e., randomly selected DNase-seq peaks in other cell types. "SS comp" and "SE comp" represent SS and SE components, respectively, throughout this figure. *P* values (**\*\****P* < 10$^{-10}$) were determined against "WG" using two-sided binomial tests without adjustment. **c** Enrichment fold of GWAS SNPs associated with individual traits. *P* values (**\*\****P* < 10$^{-10}$) were determined for significant enrichment of the traits having an enrichment fold of > 2 in a subgroup of traits as compared to all GWAS traits, using

two-sided binomial tests without adjustment ($n = 2722$ for all, 137 for immunity-associated, 8 for B-cell-lymphoma-associated, and 3 for CLL-associated GWAS traits). The center line (in red) in a box shows the median; the box bounds represent the lower and upper quartiles; the whiskers extend to the minima and maxima points up to a maximum of 1.5× the interquartile range; and dots denote outliers. **d** Fraction of replicated GWAS SNPs in different enhancer/silencer types. *P* values (**\*\****P* < 10$^{-10}$) were determined against "SS comp" (representing SS components) using one-sided binomial tests without adjustment.

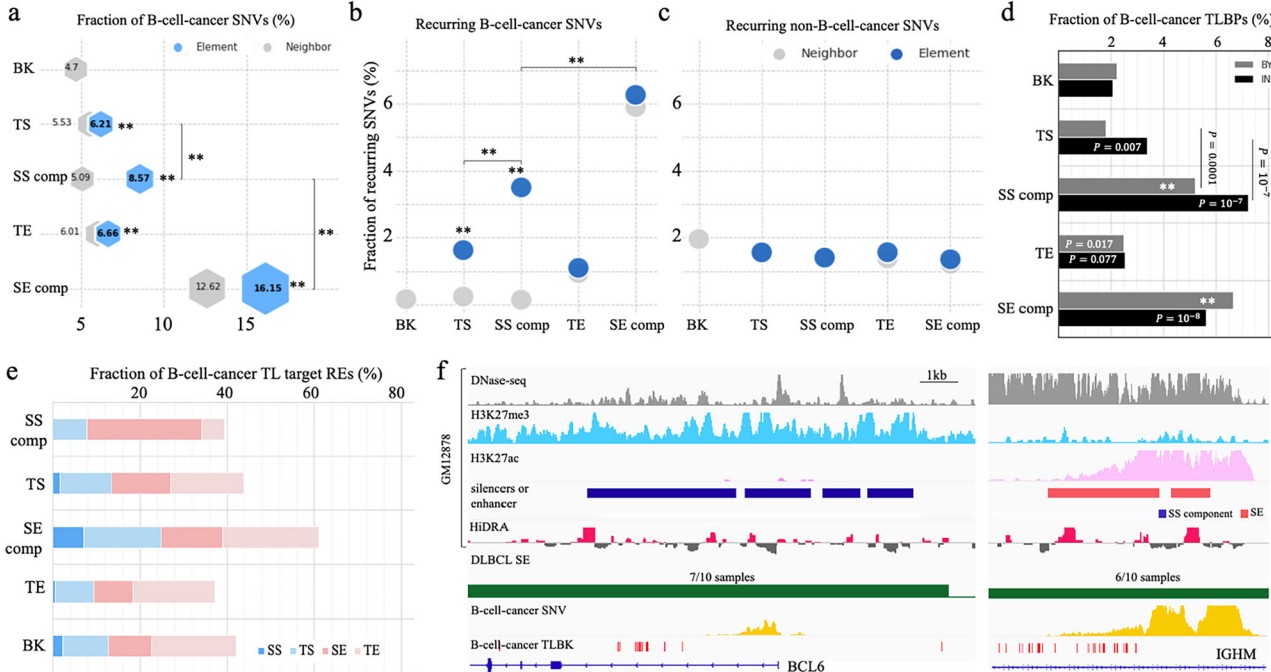

**Fig. 5 | Enrichment of B-cell-cancer variants in GM12878 SSs. a** Fractions of B-cell-cancer SNVs in different RE types. Fractions of recurrent (**b**) B-cell-cancer SNVs and (**c**) non-B-cell cancer SNVs in different enhancer and silencer types. **d** Fractions of B-cell-cancer TLBPs in and by different enhancer and silencer types. "IN" and "BY" represent the RE sequences and the genomic regions proximal to enhancers or silencers within ± 50 kb. In (**a–d**), "SS comp" and "SE comp" represent SS and SE components, respectively, while "BK" is the background sequences, i.e., randomly selected DNase-seq peaks in other cell types. In (**a, b, d**), $P$ values (** $P < 10^{-10}$) were determined (against "BK", if not specified) using two-sided binomial tests without adjustment ($n = 81,565,198$ cancer SNVs and $n = 1916$ B-cell-cancer TLBPs documented in the ICGC data portal[54]). **e** Distribution of target enhancers or silencers of

B-cell-cancer translocations (TLs). "BK" is randomly selected DNase-seq peaks in other cell types. **f** Genomic and epigenetic profiles in the BCL6 SS (left panel) and IGHM SE (right panel) regions in normal and cancer genomes. "SS comp" and "SE comp" represent SS and SE components, respectively. These two regions are merged by the DLBCL translocation t(3;14)(q27;q32). The distribution of non-B-cell-cancer mutations (i.e., SNVs and TLBPs) is not presented here since close-to-zero non-B-cell-cancer mutations are reported in these regions. Another DLBCL translocation t(3;14)(q27;p14) joins this BCL6 SS region with the RHOH SE region (Supplementary Fig. 10). The numbers in the track of "DLBCL SE" are the numbers of patients having an SE and all patients.

highlight the pivotal role of SSs in governing the fundamental activity of B cells and B-cell carcinogenesis.

## SS components host a dense population of B-cell-cancer somatic variants

We then sought to determine whether single-nucleotide variants (SNVs) in cancers show enrichment in SSs. The International Cancer Genome Consortium (ICGC) collects about 1.5 million cancer SNVs detected in two B-cell cancer types (CLL and malignant lymphoma)[54]. The density of B-cell-cancer SNVs per 1 kb in SS components is 0.50, which is the second-highest density, only after SE components (0.81), among examined silencer/enhancer sets. Further, it is significantly higher than those in the regions flanking SS components (0.30), TSs (0.38), TEs (0.33), and DNase-seq peaks randomly selected in non-GM12878 cell types (0.29) (binomial test $P < 10^{-10}$, Supplementary Fig. 12). To account for the uneven distribution of cancer SNVs in tumor genomes, we calculated the fraction of B-cell-cancer SNVs among all cancer SNVs to assess the enrichment of B-cell-cancer SNVs in SS components. We found that in SS components, B-cell cancer SNVs constituted 8.6% of all cancer SNVs, second only to SE components (16.2%), and significantly higher than TSs (6.2%) and TEs (6.7%) (binomial test $P < 10^{-10}$, Fig. 5a). In addition, B-cell-cancer SNV density in SS components is 1.7 times higher than in their flanking regions ($P < 10^{-10}$).

Next, we examined recurrent SNVs, i.e., SNVs which are detected in multiple cancer samples, as these mutations likely have a deleterious effect and can likely be considered driver mutations that confer a selective advantage to mutant cancer cells[55]. In SS components, 3.5% of

B-cell-cancer SNVs are recurrent (Fig. 5b). This frequency is comparatively lower than SE components (6.3%). Nonetheless, it is significant in two aspects: (i) it is 2 times higher than in TSs/TEs and 22 times higher than in the background (binomial test $P < 10^{-10}$); (ii) it is 2.9 times higher than that of non-B-cell-cancer SNVs in SS components ($P < 10^{-10}$, Fig. 5c). These enrichments reinforce the crucial role of SSs, together with SEs, in B cell oncogenesis.

The dense hypermutation of SSs in B-cell cancer affects master regulators of B-cell development (Supplementary Data 2). For example, the most hypermutated SS component (234 B-cell cancer SNVs per 1 kb) is close to *BCL6*, a central factor in the differentiation of normal B cells as well as malignant survival and proliferation, perhaps immune invasion, in lymphoma[56]. In addition, another two *BCL6* SS components are also hypermutated in B-cell cancers, hosting 55 and 13 SNVs per 1 kb, respectively. Furthermore, hypermutated SSs are found in proximity of *BACH2* and *CXCR4*, two TFs key for B-cell development and differentiation[36,57]. As all of these genes have been investigated as prognostic markers and treatment targets for B-cell cancers[58], the high cancer-mutation vulnerability of SS component sequences reported here could potentially provide additional insights into therapeutic strategies for designing B-cell cancer treatments.

## SS components are frequently translocated to SE regions in B-cell cancer cells

Chromosomal translocation is of importance as an oncogenic mechanism[59,60] in which nonhomologous chromosomal parts merge together, resulting in genomic fusion and disruption. As translocations directly break RE sequences and/or merge otherwise-independent

regulation domains, we analyzed the distribution of translocation breakpoints (TLBPs) within silencers or enhancers and in the regions flanking them (defined as the ±50 kb flanking regions). In general, all GM12878 silencer and enhancer sequences, as well as their proximities, are densely populated by B-cell cancer TLBPs (Fig. 5d and Supplementary Fig. 12). Among TLBPs detected in different cancer types[54] within SS components, 7.2% occur in B-cell cancers, which is higher than those in TSs (3.4%), SE components (5.6%), TEs (2.5%) and the background (2.3%; binomial test $P < 10^{-5}$ except for the comparison to that of SE components). In the proximity of SS components, the fraction of cancer TLBPs reported in B-cell cancers (5.5%) is lower than in the proximity of SE components (6.6%; $P = 0.01$), yet over 2 times greater than in the proximity of other examined regions ($P < 10^{-10}$, Fig. 5d).

Notably, B-cell-cancer translocations occurring in SS components or their proximal regions (by ±50 kb) merge SS domains to SE neighborhoods in 26% of instances, which is significantly higher than the <14% found within or proximal to the other enhancer/silencer types (binomial test $P < 10^{-10}$, Fig. 5e). These findings suggest that the juxtaposition of SSs with SEs, combining regulatory domains with strong but opposite functions, could lead to pronounced regulatory aberrations, including high upregulation of genes normally repressed by SSs (or deactivation of SE-associated genes), which then causes malignant development. For example, in 83% (14/17 reported cases) of the most well-known translocation in B-cell lymphoma, t(3;14)(q27;q32)[61], a SS region in the *BCL6* locus is brought together with a SE region in the *IGH* locus (Fig. 5f). Also, the juxtaposition of a *BCL6* SS region with an *RHOH* SE region was found in all reported cases (3/3) of translocation t(3;14)(q27;p14) (Supplementary Fig. 13). Both *IGH* and *RHOH* are highly expressed in normal B cells indicating that the merger of the *BCL6* SS domain with SE regions potentially represents a functional explanation for the aberrant upregulation of *BCL6* in B-cell lymphoma.

## Conversion of SSs to SEs underlies B-cell oncogenesis

Significant enrichment of B-cell-cancer variants (including SNVs and TLBPs) in SS components (Fig. 5, as demonstrated in the last two sections) prompted us to evaluate the altered activity of SSs during tumorigenesis in B cells. We retrieved genes experimentally verified as essential for DLBCL survival or suppressive for DLBCL proliferation[62]. Notably, GM12878 SS components are not found in any of DLBCL-proliferation suppressing loci (binomial test $P < 10^{-10}$ versus all the other tested RE types, Fig. 6a). By contrast, they are densely distributed in the loci of DLBCL-survival essential genes, with a density 3.5 times greater than the background and 1.5 times greater than the density of TSs or TEs ($P < 10^{-5}$, Fig. 6a), although this density is lower than that of SE components. These findings suggest a significant association of GM12878 SSs with loci known for the inhibition of DLBCL tumorigenesis. Although both GM12878 SEs and SSs are enriched in the loci of DLBCL-survival essential genes, they are associated with different sets of DLBCL-survival essential genes (Fig. 6b). For example, those genes uniquely associated with GM12878 SEs (such as *IRF4*, *BCL2*, *RHOA*, *PIK3R1*, *CD22*, etc.) take part in the activation of lymphocytes and B cells (as summarized in the Gene Ontology knowledgebase[63]). That is, a part of DLBCL-survival essential genes is vital for normal B cell growth and development, explaining the enrichment of GM12878 SEs in the loci of DLBCL-survival essential genes (Fig. 6a).

We compared GM12878 data with DLBCL patient data and found that 17 SS components from GM12878 fell within DLBCL-survival essential gene loci, with 11 (65%) overlapping with DLBCL SE regions detected in DLBCL patients[38]. Furthermore, of all 4,617 GM12878 SS components, 13% are converted to SE components in DLBCL patient data (denoted as SS-to-SEs below). This is significantly higher than the conversion of GM12878 TSs (7.4%) and the background sequences (2.3%; binomial test $P < 10^{-10}$, Fig. 6c). Moreover, 22% of these SS-to-

SEs conversions recur in more than half of DLBCL samples, which is significantly higher than recurrence of all DLBCL SEs (10%) and DLBCL SEs converted from GM128787 TSs (i.e., TS-to-SEs, 13%) and GM12878 TEs (i.e., TE-to-SEs, 15%, $P < 10^{-10}$, Fig. 6d). As expected, the genes proximal to SS-to-SEs exhibit the most substantial upregulation, showing an average 9.8-fold increase in expression (Student's $t$-test $p = 2 \times 10^{-7}$ vs the average 1.4-fold increase among genes proximal to all DLBCL SEs, Supplementary Fig. 14a). Significantly, among all examined regulatory elements (i.e., silencers, enhancers and CTCF binding sites) in B cells, SS-to-SEs exhibit a strikingly high density of B-cell cancer SNVs deposited in the ICGC[54]. This density is three times that of all GM12878 SSs, and 1.5 times that of GM12878 SEs ($P < 10^{-10}$, Supplementary Fig. 14b).

Consistently, 670 B cell SS components (14.5%) overlap with DLBCL SEs reported by Elodie Bal et al. [64], a fraction significantly higher than 7.9% of GM12878 TSs and 3.2% of the background sequences ($P < 10^{-10}$, Supplementary Fig. 14c). Among these 670 SS-to-SEs, 63 (9.7%) are hypermutated in multiple DLBCL samples[64], representing a twofold increase compared to the 4.3% hypermutation rate across all DLBCL SEs. DLBCL recurring SNVs in SS-to-SEs are enriched within the binding sites of TF repressors, including TGIF2, BLIMP1, NR3C1, REST, etc. (Supplementary Fig. 14d). It has been demonstrated that DLBCL recurring SNVs disrupt the binding of transcription repressors BLIMP1 or NR3C1 and thereby lead to the upregulation of SS-to-SE associated genes *CXCR4* and *BCL6*[64]. Combined, the high frequency of SS-to-SE conversions and their strong recurrence rate, likely caused by SNVs, suggest that these events represent a primary mechanism of carcinogenesis.

To investigate the impact of SS-to-SE conversions, we examined the changes in chromatin contacts linked to these regions during carcinogenesis. Approximately 9% of chromatin contacts to SS components in GM12878 cells persist in naïve Karpar-422 cells, a fraction significantly lower than the expected 12% by chance and 14% of contacts to enhancers or CTCF binding sites ($P = 0.001$, Supplementary Fig. 15a). Notably, this fraction diminishes further to 5% among chromatin interactions to SS-to-SEs. Conversely, 35% of SS-to-SE GM12878 contacts are retained in the Karpar-422 cells treated with A-485, a small molecule deactivating enhancers by inhibiting the activity of histone acetyltransferases p300 and CBP[65]. This fraction significantly exceeds the 31% of contacts to CTCF binding sites and 18% of contacts to DLBCL SEs being restored after the A-485 treatment ($P < 0.001$, Supplementary Fig. 15b). These findings suggest that SS-to-SE conversions frequently reorganize chromatin folding, a major factor contributing to gene deregulation during carcinogenesis.

We further investigated the molecular mechanism that converts GM12878 SSs to cancer SE regions. For example, the Bromodomain and extra-terminal domain (BET) protein family, specifically bromodomain-containing protein 4 (BRD4), a transcriptional and epigenetic regulator that is known for its strong enhancer binding (particularly at SEs), is often associated with the malignant proliferation of lymphoma cells[38,66]. In DLBCL Ly1 cells, 64% of SS-to-SEs are bound by BRD4, which is a significant overabundance in comparison to 56% of all DLBCL SEs ($P < 0.007$). Also, the binding intensity of BRD4 (as measured by the BRD4 ChIP-seq signal) is significantly higher in SS-to-SEs than in other DLBCL SEs, with an average 1.5-fold increase (Student's $t$-test $P = 0.002$ versus all SEs, Fig. 6e). These data are consistent with SS-to-SEs creating stronger SEs, which can accumulate more BRD4.

JQ1, a potent inhibitor of BET activity (including BRD4), effectively combats cancer by weakening enhancer activity, particularly SEs[38,67]. On average, JQ1 treatment decreases BRD4 binding intensity in DLBCL SEs by 85% in 24 h, with SS-to-SE converting regions showing the steepest decrease in magnitude (Student's $t$-test $P < 10^{-10}$ versus all SEs, Supplementary Fig. 16). Furthermore, JQ1 significantly downregulates more SS-to-SE-associated genes than genes associated with

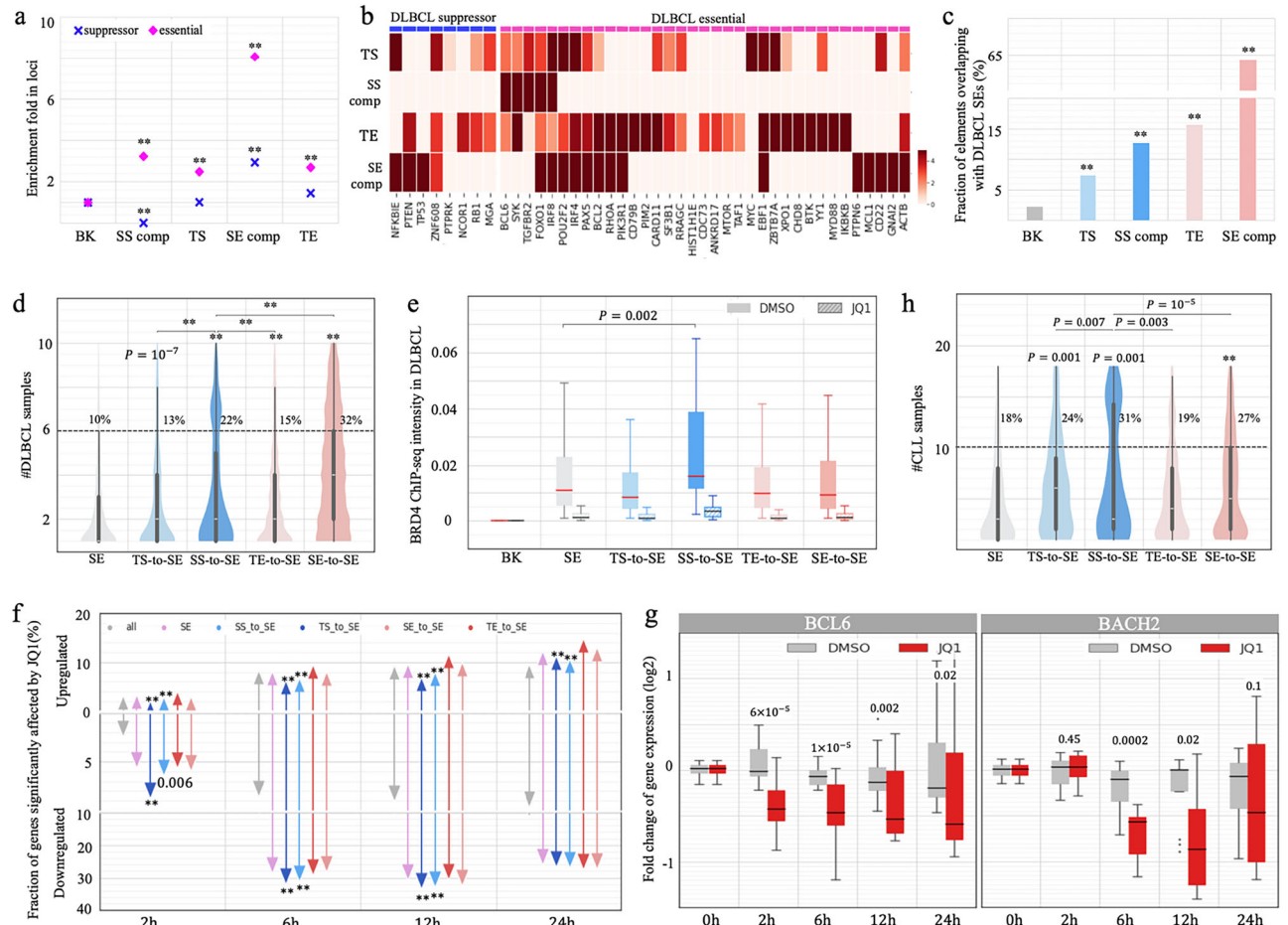

**Fig. 6 | Conversion of GM12878 SSs to SEs during carcinogenesis. a** Enrichment of SS components in the loci of DLBCL-essential and DLBCL suppressor genes. "SS comp" and "SE comp" represent SS and SE components, respectively, while "BK" is DNase-seq peaks randomly selected from other cell types, throughout this figure. **P < 10^{-10} were determined against "BK" using two-sided binomial tests without adjustment. **b** Enrichment of enhancers and silencers in each DLBCL-essential and DLBCL suppressor gene locus. **c** Fractions of GM12878 enhancers and silencers coinciding with DLBCL SE regions. **P < 10^{-10} were determined against "BK" using two-sided binomial tests without adjustment. **d** Numbers of DLBCL patients in which SEs were detected. SS-to-SE and TS-to-SE represent the DLBCL SEs converted from GM12878 SSs and TSs, respectively. P values (**P < 10^{-10}) were determined (against all DLBCL SEs if not specified) using a one-sided binomial t-test without adjustment (n = 2892). **e** Average binding intensities of BRD4 (as measured by TF ChIP-seq signals) in DLBCL SEs and BK sequences in DLBCL Ly4 cells. These intensity levels were detected at 24 h after JQ1 and DMSO treatments. The P value

was determined against Ly1-cell SEs carrying BRD4 ChIP-seq signals using a two-sided t-test without adjustment (n = 127). **f** Fractions of genes significantly upregulated and downregulated at different time points following JQ1 treatment. The genes are categorized based on their association with different DLBCL SE groups. P values (**P < 10^{-10}) were determined against all tested genes using two-sided binomial tests without adjustment (n = 28,869). **g** Expression levels of BCL2 and BACH2 at different time points following JQ1 treatment in DLBCL cells. P values between DMSO and JQ1 were determined using two-sided t tests (n = 29 biological samples). **h** Numbers of CLL patients in which SEs were detected. SS-to-SE and TS-to-SE represent CLL SEs converted from GM12878 SSs and TSs, respectively. P values (**P < 10^{-10}) were determined (against all CLL SEs if not specified) using a one-sided binomial test without adjustment (n = 1010). In (**d, e, g, h**), the center line of a box shows the median; the bounds of a box represent the lower and upper quartiles; the whiskers extend to the minima and maxima points up to a maximum of 1.5× the interquartile ranges.

all DLBCL SEs, particularly at the early time periods. For example, at 2 h after JQ1 treatment, 7.8% of SS-to-SE-associated genes are significantly downregulated, which is two times the number of all DLBCL-SE-associated genes and five times the number of all tested genes (P < 0.0001, Fig. 6f). The immediate response to JQ1 treatment can be largely attributed to the disruption of SS-to-SE function conversions by JQ1. An example of SS-to-SE-associated genes in *BCL6* illustrates this pattern. The locus hosts two BRD4-bound SS-to-SEs, in which BRD4 binding intensities are significantly higher than would be expected among all DLBCL SEs. The treatment of JQ1 reduces these by 85% after 24 h (Supplementary Fig. 16). The expression of *BCL6* significantly decreases by 14% at 2 h (Student's *t*-test P = 6 × 10^{-5}) and by 50% at 6 h (P = 10^{-5}, Fig. 6g) after the JQ1 treatment. Similarly, another DLBCL marker gene, *BACH2*, hosts two BRD4-bound SS-to-SEs in its locus and is significantly downregulated by JQ1 treatment (P = 0.0002 at 6 h,

Fig. 6g). These results highlight the crucial role of BET in SS-to-SE conversion, leading to dysregulation of associated genes during DLBCL oncogenesis. However, this may not explain the whole mechanism of converting normal-B-cell SSs to cancer SEs, as BET is codependent with other co-factors for transcriptional activation[68]. Further examination in other co-factors and TFs will help understanding how normal-cell SSs are converted to SEs during carcinogenesis.

Similarly, we checked data from CLL samples, obtaining 1010 distinct CLL SE regions[69]. There is a significant enrichment of GM12878 SS components in CLL SE regions (94 out of 4,617 or 2.04%) compared to TSs (1.11%) and to background (0.26%, P < 10^{-10}; Supplementary Fig. 17a). Moreover, the CLL SS-to-SE conversions that recur in more than half of samples (31%) are significantly greater than TS-to-SEs recurrences in CLL (24%) and CLL SEs (18%, P < 10^{-3}; Supplementary

Fig. 17b). These results, which are consistent with those in DLBCL samples, further confirm our conclusion that the conversion of SSs to SEs play a major factor of CLL carcinogenesis.

In addition, we investigated the conversion of SEs to SSs, which represents the extreme opposite of the SS-to-SE conversion, during B cell carcinogenesis (Supplementary Information). Of GM12878 SE components, 4.8% function as a SS in at least one examined B-cell cancer cell type, significantly exceeding 4.3% observed in the background regions or TSs (binomial test $P \leq 0.01$, Supplementary Fig. 18a). Furthermore, 7.4% of these SE-to-SSs recur, exceeding the 5.0% recurrence rate among the background sequences ($P < 10^{-10}$, Supplementary Fig. 18b). Furthermore, SE-to-SSs exhibit greater loss in chromatin contacts than the other SEs ($P = 0.0001$, Supplementary Fig. 15a). These findings, along with the high recurrence of SS-to-SE conversions (Fig. 6), suggest that 1) large-scale functional overturns occur during DLBCL development; and 2) these overturns often coincide with chromatin reorganizations. However, unlike SS-to-SEs, which typically harbor an exceptional density of cancer SNVs, SE-to-SSs exhibit a density of B-cell lymphoma SNVs similar to other GM12878 SEs (Supplementary Fig. 14b). These results suggest that SE-to-SS conversions are less likely to play a dominant role in B cell lymphoma.

It is worth noting that the large fraction of GM12878 SE components overlapping with cancer SE regions (Fig. 6c and Supplementary Fig. 17) and the high recurrence rate of these cancer SE regions (Fig. 6d, h) reflect that some genes (and their transcriptional activation mechanisms) are essential for the survival of both normal and malignant B cells, as discussed earlier (Fig. 6a, b).

## SSs are enriched at TAD shores and associated with TAD formation

As H3K27me3/Polycomb acts as a mediator of high-order chromatin organization[70,71], we investigated the contribution of SSs to TAD formation and maintenance in GM12878 cells. SSs are located closer to GM12878 TAD-boundaries than the other silencer/enhancer types, although further away than ChIP-seq binding sites of CTCF, a well-known chromatin structural regulator (Wilcoxon rank sum test SS component versus TS $P = 0.01$; versus SE component and TE $P < 10^{-10}$, Supplementary Fig. 19). Specifically, SSs are located preferentially at GM12878 TAD-shores (defined as the regions 20 kb to 50 kb from a TAD-boundary) but not in flanking TAD-boundary regions (defined as the regions in ≤20 kb from a TAD-boundary). That is, more SS components (10%) reside at TAD-shores as opposed to TSs (7%), TEs (6%), and SE components (5%; binomial test $p \leq 0.03$, Fig. 7a).

To further characterize the function of TAD-shore SSs, we evaluated the density of silencers and enhancers in the SS TAD-shores (i.e., GM12878 TAD-shores hosting at least one SS component) and TAD-boundaries. As expected, these regions are depleted of TEs and SEs (Student's t-test $P \leq 0.05$, Fig. 7b, c), likely due to the spread of strong H3K27me3 in SSs inhibiting the deposition of H3K27ac[72]. Also, SS-counterpart-TAD-shores (defined as the TAD-shores on the side of a TAD-boundary opposite to a SS-TAD-shore) host more TEs and SE components than all TAD-shores or CTCF-counterpart-TAD-shores (i.e., the TAD shores on the side of TAD boundaries opposite to a CTCF TAD-shore, Student's t-test $P \leq 0.05$, Fig. 7c).

To address a hypothesis that TAD-shore SSs might be establishing a transcriptional boundary, which prevents propagation of SE-based activation, we computed the density of B-cell-cancer TLBPs at TAD-

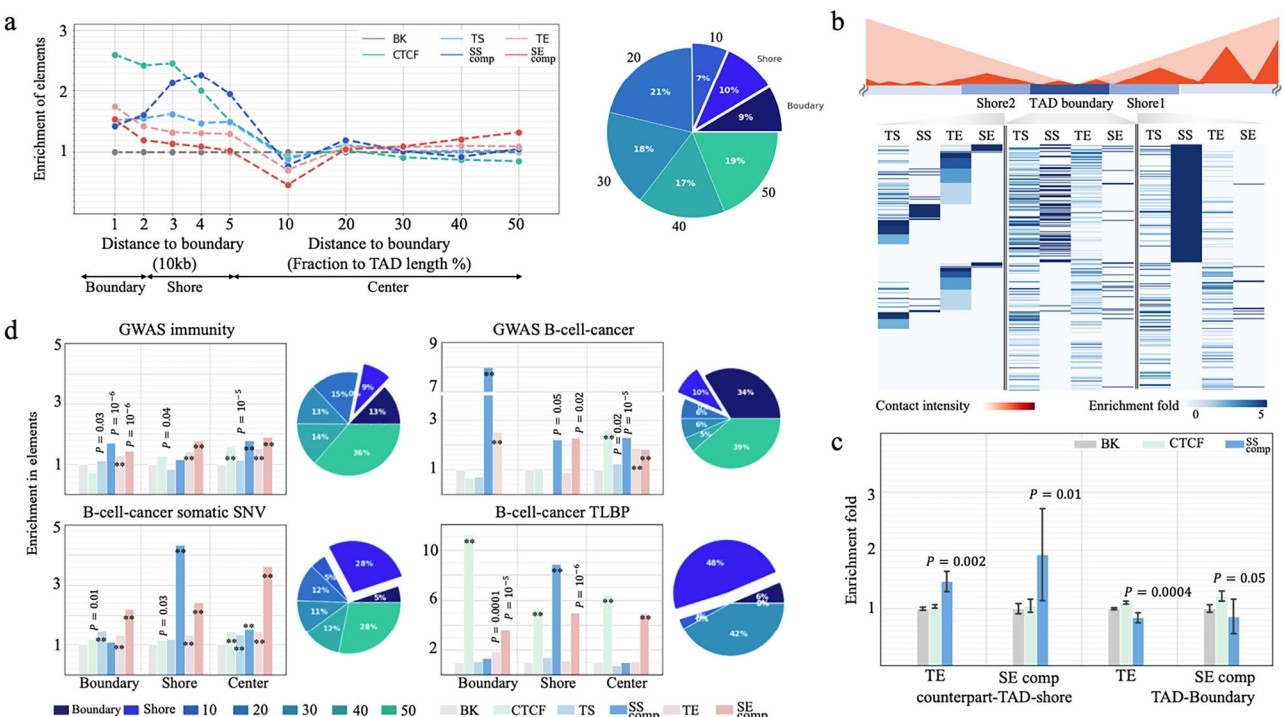

**Fig. 7 | Enrichment of SSs in TAD-shores. a** Distribution of REs in different TAD sections (boundaries, shores, and centers). The pi chart presents the distribution of SSs in TADs. "SS comp" and "SE comp" represent SS and SE components, respectively, throughout this figure. **b** Enrichment of silencers and enhancers in SS-counterpart-TAD-shores. The schematic shows two neighboring TADs. The Shore2 is the counterpart-TAD-shore of the Shore1, and vice versa. Each row in this heatmap represent a TAD boundary. Enrichment fold is calculated through comparing with the whole genome. **c** Enrichment of enhancers in the SS-counterpart-TAD-shores. CTCF-TAD-shores have no significant preference for enhancers. Data are presented as the median±SEM. $P$ values (**$P < 10^{-10}$) were determined against all TAD boundaries in GM12878 ($n = 6240$ as documented in the Peakachu[31]) using two-sided Wilcoxon rank sum tests without adjustment. **d** Enrichment of disease-associated mutations in different RE types across TAD sections. Pie charts are the distribution of mutations in SS components. "BK" is DNase-seq peaks randomly selected in other cell types. $P$ values (**$P < 10^{-10}$) were determined against "BK" using one-sided binomial tests without adjustment.

shore SSs. As cancer TLBPs are known to frequently undermine TAD formation[73,74], their enrichment at TAD-shore SSs should be indicative of the TAD-shore SS in TAD formation. Consistent with our hypothesis, the density of B-cell-cancer TLBPs in TAD-shore SS components is 8.9 times greater than in the whole genome and 6.1 times greater than in non-TAD-shore SS components ($P < 10^{-10}$). Furthermore, TAD-shore SS components constitute 10% of all SS components, yet they host 48% of SS B-cell-cancer TLBPs ($P = 10^{-42}$, Fig. 7d). These findings indicate that TAD-shore SSs, like CTCF binding sites[75], are strongly linked with TAD maintenance, and possibly formation.

To account for the biased distribution of cancer mutations, we also used the fraction of B-cell-cancer mutations among all cancer mutations to evaluate the enrichment of B-cell-cancer variants across enhancer and silencer sets. The fraction-based enrichment of B-cell-cancer TLBPs remains at an exceptionally high level of 7.9-fold in TAD-shore SS components (binomial test $P < 10^{-10}$) but greatly decreases to 1.9-fold in boundary CTCF binding sites (Supplementary Fig. 20). These results validate a cell-specific role of GM12878 SS components in TAD-boundary formation and also reflect the conservation of CTCF binding profiles across tissues/cell types[76].

Furthermore, B-cell cancer GWAS SNPs are enriched in SSs across all TAD segments, which likely reflects the multi-component nature of SSs and susceptibility of multiple SS parts to mutations. Contrasting this observation with the selective B-cell cancer TLBP enrichment in TAD-shore SSs, we conclude that large-scale chromosomal rearrangements originating at TAD-shore SSs are likely to remove large parts of SS, resulting in the most pronounced phenotypic impact and, thus, leading to carcinogenesis. Effectively, TAD-

shore SS TLBPs are acting as a set of multiple mutations affecting individual SS parts simultaneously (Fig. 7d). This results in different carcinogenesis pathways driven by germline mutations and somatic translocations, similarly to what has been reported for protein-coding regions[77]. Notably, GWAS B-cell-cancer SNPs are frequently located in the loci of *ACOLX, BCL6, GATA3, POMC, RBM38*, etc., while B-cell-cancer TLBPs are predominantly enriched in the SSs proximal to BCL6.

## CpG islands and silencing mechanisms of SSs
Next, we examined chromatin contacts of SS components. As expected, all silencers and enhancers have more GM12878 chromatin contacts than randomly selected DNase-seq peaks in other cell types (Student's *t*-test $P < 10^{-5}$, Fig. 8a), reflecting dense chromatin contacts to active silencers or enhancers. Also, silencers and enhancers of the same type often interact with each other. For example, SS components interact with the other SS components 2.5 times more than with TSs and 3.5 times more than with enhancers (Student's *t*-test $P < 10^{-5}$), as also reported on MRR in the Cai et al. study[25].

Similarly, SE components interact most frequently with other SE components ($P < 10^{-5}$ versus TEs and silencers). Given the significant enrichment of CGIs in SS components (Fig. 1c) and an elevated number of chromatin contacts involving CGIs, including promoter and enhancer CGIs[78], we subcategorized SS components into those overlapping with CGIs by over 200 bps (named CGI SS components) and those with no CGI overlaps (named non-CGI SS components). Our results show that CGI SS components feature significantly more chromatin contacts than non-CGI SS components ($P = 10^{-6}$, Fig. 8a).

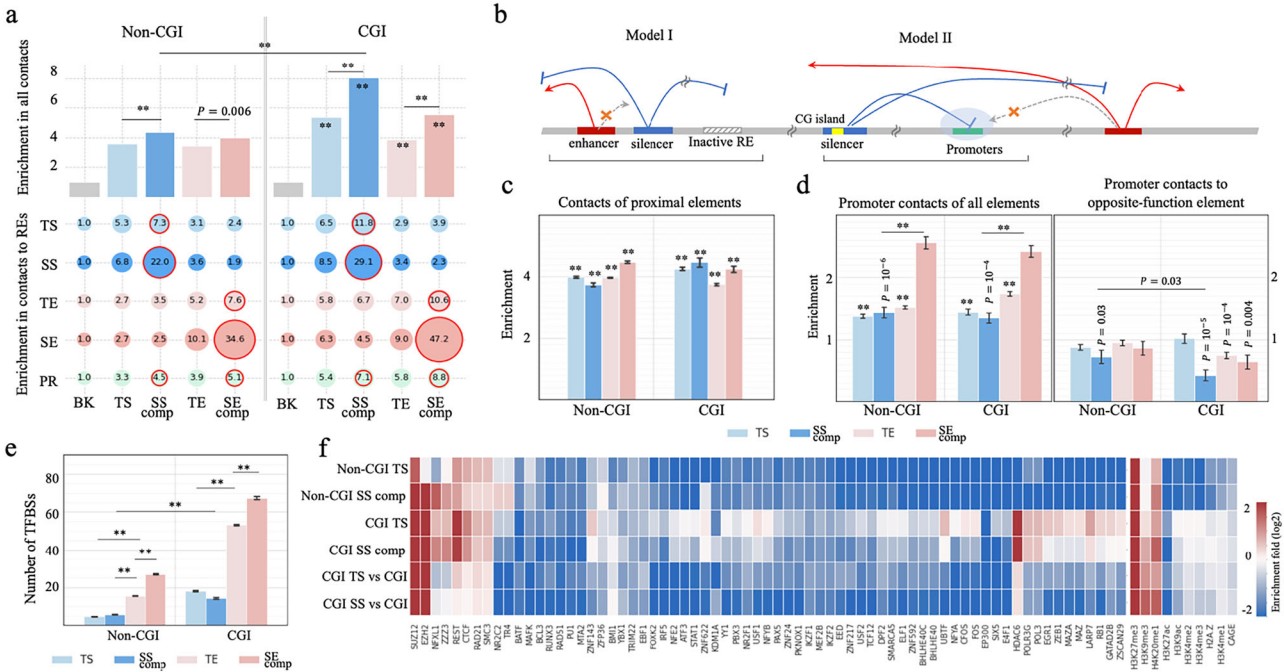

**Fig. 8 | Repression models of SSs. a** Enrichment of chromatin contacts to different enhancer and silencer types (as illustrated in bar plots) and contact enrichment between two enhancer and silencer types (in the bubble plots). Throughout the figure, "SS comp" and "SE comp" represent SS and SE components, respectively. *P* values (**$P < 10^{-10}$) were determined (against "BK", i.e., the background sequences, which were randomly selected DNase-seq peaks in other cell types, if not specified) using one-sided binomial tests without adjustment. "PR" represents all protein-coding promoters. **b** Two major repression models utilized by silencers. **c** Contact enrichment in the proximities of different enhancer and silencer types. Data are presented as the median±SEM. *P* values (**$P < 10^{-10}$) above bars were determined against all chromatin contact anchors in GM12878 (as documented in the Peakachu[31]) using two-sided *t* tests without adjustment (*n* = 22,817). **d** Contact

enrichment of promoters, which are categorized based on their enhancer or silencer contacts. Data are presented as the median±SEM. *P* values (**$P < 10^{-10}$) above bars were determined against all promoters having Hi-C contact in GM12878 (as reported in the Peakachu[31]) using two-tailed *t* tests without adjustment (*n* = 6951). **e** Numbers of ChIP-seq TFBSs located within silencers and enhancers. Data are presented as the median±SEM. **$P < 10^{-10}$ were determined by two-sided *t* tests without adjustment (*n* = 2,755,484 ChIP-seq peaks for 168 TFs or co-transcription factors, see Supplementary Information). **f** Signatures of ChIP-seq TFBSs and epigenetic marks in CGI and non-CGI silencers. All enhancers were used as the background in the top four rows. CGI enhancers were used as background for the bottom two rows. High similarity between the top two and bottom two rows suggests that CGI is one of the factors determining TFBS features of CGI silencers.

Two main models summarize the repression mechanisms of silencers (Fig. 8b). Model I is characterized by silencers condensing local genomic regions, dampening chromatin contacts of proximal REs. Model II involves silencers occupying gene promoters, shielding them from enhancer interactions (Fig. 8b)[79]. To understand the relevance of these two models to SSs, we evaluated chromatin contact enrichment near SS components within ±50 kb flanking regions. The proximity of non-CGI SS components has the lowest contact density compared to other non-CGI silencer and enhancer types (Student's $t$-test $P < 0.0007$), while CGI-SS-component proximity features the highest contact density among all CGI-silencers and enhancers ($P \leq 0.07$, Fig. 8c). Therefore, non-CGI SSs more frequently use model I than CGI SSs. In addition, although the promoters targeted by SS components have more chromatin interactions than randomly selected promoters ($P < 10^{-10}$, the left panel of the Fig. 8d), they lack enhancer interactions ($P = 10^{-5}$, the right panel of the Fig. 8d). Notably, the promoters targeted by CGI SS components have the least enhancer contacts compared to the promoters targeted by enhancer and TSs, including those targeted by non-CGI SS components ($P = 0.03$). In conclusion, both repression models are utilized, but CGI and non-CGI SS components have different preferences in repressive mechanisms. CGI SS components, which have dense contacts on their own (Fig. 8a) and in their proximity (Fig. 8c), are biased toward model II, shielding their target promoters from enhancer contacts (Fig. 8d). Non-CGI SS components often use model I, reducing chromatin contacts in their proximity. These trends were also observed in other cell types, including H1 hESC, HepG2 and K562. Specifically, CGI SSs and their adjacent regions have more Hi-C contacts than non-CGI SS counterparts ($P < 10^{-10}$ in most of the comparisons, the plots in the left column in Supplementary Fig. 21). On the other hand, the promoters linked with CGI SSs show increased densities of total contacts but decreased densities of enhancer contacts ($P < 10^{-10}$, the plots in the middle and right columns, Supplementary Fig. 21). It is important to note that these models can be further explored in detail, especially with high-resolution and short-distance contacts on a large scale (chromatin contacts used here are in a resolution of 10 kb and in lengths of >80 kb[31]). Moreover, the gene promoters targeted by SSs show a reduction in enhancer (including SE) contacts (Fig. 8d). Despite their scarcity, it is worth mentioning that 12 gene promoters are linked to both SSs and SEs. Among these genes are RARA, NEBAL2, ZNF358, and ATG5, all of which contribute to immune system development and hematopoietic organ development[80].

We compared CGI and non-CGI SS components in terms of ChIP-seq TFBSs. Silencers are less frequently bound by TFs than enhancers (Student's $t$-test $P < 10^{-10}$, Fig. 8e). This indicates that silencers could be more selective for bound TFs than enhancers[81]. However, this could also be partially due to fewer repressors than activator TFs being profiled by TF ChIP-seq experiments. Furthermore, all silencers, regardless of the degrees of their overlap with CGIs, including SS components and TSs, are enriched in the TFBSs of the repressors REST and CTCF (binomial test $P < 10^{-5}$ versus all enhancers), while they are depleted of ChIP-seq peaks of the co-activator EP300 ($P < 10^{-5}$, Fig. 8f). Moreover, repressive marks H4K20me1 and H3K9me3 are enriched in SS components and TSs ($P < 10^{-5}$).

CGI SS components and CGI TSs host more TFBSs than non-CGI counterparts (Fig. 8e). For example, both CGI SS component and CGI TSs are enriched in the TF ChIP-seq peaks of HDAC6, a histone deacetylating factor responsible for epigenetic repression. This suggests that CGIs in silencers are one of the factors dictating which repressive TFs and regulation circuits are utilized by silencers.

The difference between CGI and non-CGI SS components in terms of silencing model preferences and TFBS enrichment led us to compare their functional impact. Both subclasses of SS components are densely distributed in TAD-shores and are significantly enriched in disease-associated mutations (including GWAS SNPs associated with immune systems or B-cell-cancer-associated, and B-cell-cancer somatic variants, Supplementary Fig. 22). This suggests both CGI and non-CGI SSs are of functional importance, even though they are potentially bound by different TFs and employ different silencing mechanisms.

## Discussion

Silencers have been observed to form genomic regions that are characterized by exceptionally high H3K27me3 signals and notable repressive influence in gene regulation[24,25]. In the K562 cell line, the knockout of the active region of a SS (or referring to as an H3K27me3-rich region) results in the upregulation of the target genes, including FGF18, a gene involved in cell differentiation and cell migration, and subsequently inhibits tumor growth[24,25]. In this study, we have identified SSs in different cell lines as the regions having the densest concentrations of silencers and carrying significant H3K27me3 signals. The predicted SSs are consistently linked to the genes having the lowest expression levels, aligning with observations in Cai's study[25]. We further showed that, in differentiated B cells, SSs downregulate genes highly expressed during development, especially in early developmental stages. These SSs are notably enriched in gene loci essential to B cell differentiation and development, including BCL6 and FOXO1, while being absent in the tumor supressor genes, such as TP53 and NCOR1. This suggests that SSs may be essential in defining cell identity by suppressing development-specific genes. SSs, much like SEs among enhancers, have a particular significance among silencers. In addition, we found a higher concentration of immune-system-associated SNPs and B-cell-cancer variants within SS components, exceeding those in TSs and TEs, and in some instances rivaling SE components.

To elucidate the role of SSs in the maintenance of cell identity, we explored their association with chromatin architecture, considering that 1) 3-D chromatin folding is crucial for establishing and maintaining cell identity[82,83]; and 2) H3K27me3 serves as a mediator of chromatin architecture[84]. In the K562 cell type, deactivating an SS results in the loss of the local TAD and chromatin loops[24]. Our study demonstrates the abundance of SS components in proximity to TAD boundaries. These TAD-boundary-proximal SS components are uniquely enriched in B-cell cancer TLBPs, second only to TAD-boundary binding sites of CTCF, the most well-known 3D-genome regulator. In addition, during the carcinogenesis of B-cell lymphoma, SS components exhibit more loss in chromatin contacts than TSs, enhancers, and CTCF binding sites. Collectively, our findings strongly support the pivotal role of SSs in shaping and maintaining the genome's TAD architecture. Despite these insights, the molecular mechanisms through which SSs contribute to the establishment and maintenance of TADs and chromatin loops are not entirely clear, and further research is needed to uncover the specifics. Notably, the SS components proximal to TAD boundaries show a similar TFBS profile, epigenetic signature, and CGI overlap with the other SS components, leaving it unclear how they strengthen TAD boundaries, if they directly contact TAD boundaries and/or if they recruit molecular components essential for TAD formation.

Emerging evidence indicates that cancer progression is associated with the activation or acquisition of SEs[85,86]. Interestingly, our study discovered that the formation of cancer SEs often involves the functional conversion of silencers, especially SSs. Although rarely explored, if at all, the conversions from SSs to cancer SEs occur on a large scale. For example, 13% of SS components in the GM12878 cell line convert into DLBCL SE regions, including those found within key DLBCL regulatory genes, such as BCL6 and BACH2. Importantly, the genomic regions undergoing these conversions are associated with remarkable upregulation of gene transcription. They exhibit the strikingly high density of cancer somatic variants in the case of B-cell cancers. Furthermore, these regions display the most significant loss in chromatin contacts in naïve DLBCL cells and the most significant re-establishment in chromatin structure after deactivating enhancers in DLCBL cells, among all examined regulatory elements, including CTCF binding

sites. Our findings repeatedly underscore the crucial contribution of the conversions from SSs to cancer SEs to carcinogenesis, partly through reorganizing the 3-D chromatin structure, a well-known underpinning of carcinogenesis[83]. Further investigations on SS-to-SE conversions, probably the reversal of these conversions, could potentially provide promising hints for cancer therapy.

As demonstrated in this study for primary B cells and supported by previous research, the GM12878 cell line effectively recapitulates the cellular behaviors and molecular characteristics of primary B cells[39–42]. Its vast sequencing data resources and ease of targeted modifications further establish its widespread use as a surrogate for primary B cells. However, experimental validation in primary B cells and in vivo conditions may be necessary to ensure the broader applicability of our findings to different B cell subtypes across varied cellular contexts. We note that follow-up studies of super-silencer activity may benefit from the use of episomal vectors in cycling cells, which would ensure that silencing is assessed within a dynamic, chromatinized environment and offer an additional measure of the repressive effects exerted by super-silencers.

Unlike SEs, which are densely populated by Mediator coactivators[23], SSs and TSs show similar TFBS enrichment. This might suggest either an absence of SS-specific factors or that further genomic and epigenetic exploration is required to identify these unique elements. To conclude, our study underscores the vital role SSs play in shaping cellular identity and managing core cellular processes. The significant enrichment of cancer-associated mutations within SS sequences also points to the potential of SSs as promising therapeutic targets for not only cancer but also other complex diseases.

## Methods

Research performed in this study complies with all relevant ethical regulations. This study complies with all relevant ethical regulations as outlined and approved by the institutional review board of the National Institutes of Health (NIH), USA (under NIH Clinical Center Institutional Review Board-approved protocol 99-GC-0168). Written informed consent was obtained from all study participants.

### Prediction of SSs

We used the deep learning classifier built in the previous study[87] to screen, in H3K27me3 ChIP-seq peaks, the active segments (i.e., carrying at least one DNase-seq peak or TF ChIP-seq peak) with a 1 kb window and a step of 100 bp. Each 1-kb-long input sequence was thus evaluated by a silencer prediction score. The cutoff for labeling silencers (say $t_s$) was set as a false positive rate (FPR) of 0.1 in test samples, with the size ratio of background samples to candidate silencers of 9:1. We identified 34,605 silencers by collecting the sequences having a silencer prediction score larger than $t_s$ and merging the overlapping sequences. Similarly, we screened DNase-seq and TF ChIP-seq peaks located within H3K27ac ChIP-seq peaks and identified 61,527 enhancers based on enhancer prediction scores. The average lengths of identified silencers and enhancers are 1492 bps and 1593 bps, respectively.

The Rank Ordering of Super-Enhancers (ROSE) program[23] was applied to these predicted silencers or enhancers for the identification of SS or SE regions, i.e., the genomic regions harboring a cluster of silencers/enhancers and exhibiting a high level of H3K27me3/H3K27ac modification. All files used here were downloaded from the Roadmap Epigenomic Project[21] and the Encyclopedia of DNA Element (see Supplementary Information)[88].

### Experimental test of SSs

To test the function of SSs, we selected two silencer components (i.e., chr6:90994680-90995880 and chr6:91004111-91006111) located next to BACH2 gene, as BACH2 is a master gene for the development of B-cells and for oncogenesis of B-cell cancers. As luciferase assays

cannot measure genomic sequences >500 bp, we concentrated on smaller, potentially functional sections in the two SS components. We determined functional sections of DNase-seq peaks, which are known to mark open chromatin regions (OCRs) accessible for the binding of TFs and other regulatory factors. This yielded four fragments, i.e., SS1, SS2, SS3, and SS4 (Fig. 2a). These fragments are evolutionarily conserved between humans and mice[89]. We also selected three control sequences in the BACH2 locus, either bordered by a tested SS component (i.e., SSctrl1 and SSctrl2) or active in GM12878 (i.e., SSctrl3, a proximal DNase-seq peak not identified as a silencer, Supplementary Data 3). We also extracted six SS segments in the BCL6 locus (Supplementary Data 3). Similarly, these segments are evolutionarily conserved between humans and mice. These sequence fragments were synthesized by Thermo Fisher Scientific (Waltham, MA) and cloned into plasmid DNAs in the upstream of an enhancer, SV40 promoter, and luciferase reporter gene using the Invitrogen Gateway Cloning System. Cell transfection and expression measurement were identical to those previously published[90].

### Primary B cell isolation

Peripheral blood mononuclear cells (PBMCs) were obtained by density-gradient centrifugation from healthy individuals enrolled under NIH Clinical Center Institutional Review Board-approved protocol 99-GC-0168. B cells were isolated from PBMCs by negative magnetic bead-based selection using a B cell enrichment cocktail (StemCell Technologies). The purity of the B cells was greater than 95%.

### Cooperativity among silencers and enhancers

To evaluate cooperativity among silencers and enhancers computationally, we resorted to their potential activity across cell lines, assuming that enhancers or silencers closely cooperate when their activities across cell lines are positively correlated. In any given cell type, genomic segments overlapping with H3K27ac and H3K27me3 ChIP-seq peaks were labeled as 1 and -1, respectively. The segments having both H3K27ac and H3K27me3 signals were regarded as uncertain and thus marked with 0. The cooperativity between two elements (say $i$ and $j$) was then evaluated with the cosine similarity, that is,

$$Cooperativity_{ij} = \frac{\sum_{k \in K} a_{ik} a_{jk}}{\sqrt[2]{\sum_{k \in K} a_{ik}^2}\sqrt[2]{\sum_{k \in K} a_{jk}^2}} \tag{1}$$

where $a_{ik}$ and $a_{jk}$ are the activity of the elements $i$ and $j$ in a cell type $k$, respectively. $K$ is a set of the tested cell types. The higher the value of $Cooperativity_{ij}$, the more likely it is that $i$ and $j$ have synchronized activity. Furthermore, we evaluated all pairwise cooperativity among SS components within the same SS regions and compared it with cooperative values of TS pairs that were randomly selected to have in-between distances matching the distances among SS components.

TFBS similarity between two SS components was estimated with Jaccard similarity. That is,

$$SIM_{TFBS} = \frac{\sum_{g \in G} t_{ig} t_{jg}}{\sum_{g \in G} \left( t_{ig} + t_{jg} - t_{ig}{}^* t_{jg} \right)}, \tag{2}$$

where $t_{ig}$, a binary variable, encodes the presence (1) or absence (0) of the binding site of TF $g$ in the element $i$. $G$ is a set of TFs. TFBSs here are profiled based on TF ChIP-seq peaks. The ChIP-seq data for 168 TFs/mediators in GM12878 were downloaded from the Encyclopedia of DNA Element[88].

### GWAS SNPs

We downloaded the GWAS SNPs curated and assembled into the National Human Genome Research Institute (NHGRI) catalog as of

January 2019[48]. To account for the fact that GWAS SNPs might not be directly responsible for phenotypic alterations but might be in a linkage disequilibrium (LD) with untested causal genetic variants[91], we further expanded the SNP set by adding SNPs in a strong LD ($r^2 > 0.8$) block with GWAS SNPs in at least one population of the 1000 Genomes Project.

## Reporting summary

Further information on research design is available in the Nature Portfolio Reporting Summary linked to this article.

## Data availability

H3K27ac and H3K27me3 ChIP-seq broad peaks and DNase-seq peaks of the GM12878 cell line were downloaded from the Roadmap Epigenomics Project (http://egg2.wustl.edu/roadmap/data/ byFileType/peaks/consolidated/broadPeak/). 168 TF ChIP-seq peaks for the GM12878 lymphoblastoid cell line as of April 2021 from the Encyclopedia of DNA elements project (https://www.encodeproject.org/)[92]. Cancer SNVs and TLBPs were downloaded from the ICGC data portal as of November 2019[54]. GM12878 methylation data were retrieved from GEO accession GSE155791[93]. Chromatin contacts and TADs were predicted with the framework Peakachu[31]. DLBLC data (including SEs, BRD4 ChIP-seq signals and peaks, gene expression profiles) were downloaded from GEO accession GSE45630 and GSE46663[38]. ACAT-STARR-seq activity scores were downloaded from GEO accession GSE181317[33]. High-resolution dissection of regulatory activity screening (HiDRA) was downloaded from the GEO association GSE104001[94]. H3K27ac and H3K27me3 ChIP-seq data of primary B cells were downloaded from the ENCODE project (biosample ID CL:0000788, https://www.encodeproject.org/biosamples/ ENCBS857XIR/)[92]. H3K27ac and H3K27me3 data of DLBCL cell lines, including Ly1 (ENCODE biosample ID EFO:0005907), Ly3 (EFO:0006710), Karpas-422 (EFO:0005719), and SU-DHL-6 (EFO:0002357), were downloaded from the ENCODE project[92]. DLBLC data (including SEs, BRD4 ChIP-seq signals and peaks, gene expression profiles) were downloaded from GEO accession GSE45630 and GSE46663[38]. DLBCL SEs were detected in two independent studies, one by Chapuy et al.[38] and the other by Bal et al.[64]. The former one involved 4 DLBCL patients and 6 DLBCL cell lines, while the latter one investigated 29 DLBCL cell lines. CLL SEs were detected from 18 patients by Ott et al.[69]. Information of these samples is provided in Supplementary Data 4. Silencers (including SSs and TSs) predicted in GM12878 and primary B cells were listed in Supplementary Data 1 and Data 5. Luciferase experiment results were listed in Supplementary Data 3.

## Code availability

The source code (building the CNN model and predicting enhancers and silencers) is available on GitHub (https://github.com/ncbi/SilencerEnhancerPredict). Data for training (and predicted silencers and enhancers) are available at https://zenodo.org/records/16241561.

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

## Acknowledgments

This research was supported by the Intramural Research Program of the National Library of Medicine (NLM) (ZIA LM200881 to IO), the National Human Genome Research Institute (NHGRI) (ZIA HG200323 to LE), the National Institute of Allergy and Infectious Diseases (NIAID) (ZIA AI000825 to SM) and the National Institute of Diabetes and Digestive and Kidney Diseases (NIDDK) (ZIA DK075149 to BA) within the National Institutes of Health (NIH). The contributions of the NIH authors were made as part of their official duties as NIH federal employees, are in compliance with agency policy requirements, and are considered works of the United States government. However, the findings and conclusions presented in this paper are those of the authors and do not necessarily reflect the views of the NIH or the U.S. Department of Health and Human Services.

## Author contributions

D.H. performed computational analysis and analyzed the data. H.M.P., L.E., D.K., L.K., S.M., and B.A. performed experimental validation of silencers and analyzed experimental results. I.O. supervised computational work. L.E., B.A., and S.M. supervised the experimental component of the study. D.H. prepared figures and tables. D.H., B.A., L.E., and I.O. wrote the manuscript.

## Funding

## Competing interests

The authors declare no competing interests.
