## [Peer Review file · Nature Communications]

Super-silencers are crucial for development and carcinogenesis in B cells.

Corresponding Author: Dr Ivan Ovcharenko

Version 0:

Reviewer comments:

Reviewer #1

(Remarks to the Author)

In this manuscript, Huang et al investigate putative supersilencers and their roles and potential mechanisms in GM12878 and B cell lymphoma cell line data. While the analyses shown here largely make sense, and there is great interest in the growing field of supersilencers, there is very little wet lab experimental testing, meaning that many conclusions are not rigorously supported. Additionally, a lot is over-claimed on the basis of GM12878 and a few cell lines, but usually to make claims about cancer, more clinical samples (both cancer and normal) are needed.

Other major comments

1. The two models (Figure 8) are interesting but there is insufficient experimental evidence provided for them. Ideally multiple lines of experimental evidence should support these two models
2. Only GM12878 is used as a "normal" B cell. It is not even very normal as it grows in culture and is EBV infected. More experimental data from normal B cells should be given.
3. More B cell lymphoma results should be included. Also, are there controls for batch effects because these different datasets may have come from different sources?
4. Clinical B cell lymphoma results should also be given. The data downloaded from GEO comes from just a few cell lines, as far as I am aware.
5. The title is too broad considering the results shown - not all of carcinogenesis is shown. Only B-cell lymphoma is discussed.
6. Luciferase reporter is an in vitro assay and somewhat artificial. More could be done (eg CRISPR excision or modulation) for more biologically relevant results.

Reviewer #2

(Remarks to the Author)

Super silencer (SS) is an emerging concept that describes genomic regions with the most density in histone repressive marker (H3K27me3). The authors performed a comprehensive analysis on SS in B cell lymphoma using public data generated for GM12878. The analysis showed that SSs are strongly associated with developmental-specific genes, enriched with cancer-specific SNPs/variants, and linked to conversion to cancer-specific super enhancers. The authors further found that SS are enriched at TAD shores unique to B-cell cancers, supporting a potential role in sharpening the 3D chromatin architecture in cancer cells. Generally, the experiments are well designed, the data analysis is well controlled, and the results are supported by the analysis. I have two points for the authors to consider.

- 1) A clear distinguishing of the current manuscript from two previous manuscripts (refs 24 and 25) should be fully discussed in the discussion section.
- 2) The analysis on transcription response to JQ1 for SS-associated genes could be extended to the context of 3D chromatin. It is not clear whether the "SS-to-SE" genes include those identified through 3D chromatin-chromatin interaction or not. If not, that will be something to enhance the conclusions of the manuscript for the role of SS in modulating TAD structures.

Reviewer #3

(Remarks to the Author)

The study explores variations in the repressive H3K27me3 signals in silencers, emphasizing 'super-silencers.' These regions were initially linked to gene upregulation during development and impact B-cell function. Some super-silencers transform into super-enhancers in B-cell lymphoma, with key genes responding more rapidly to cancer therapy. Super-silencers also exhibit an overabundance of B-cell cancer-related mutations and translocation breakpoints, underscoring their pivotal role in B-cell regulation. The CpG content in super-silencers influences their repressive mechanisms, with implications for B-cell carcinogenesis.

Major Comments:

The author predicted SSs and SE in the GM12878 cell line and shows low DNA methylation levels, CTCF binding, and TADs among all those regions. It would be interesting to check gene deserts and housekeeping genes among those regions.

The author mentioned testing silencing activity using luciferase reporter assays. And none of the examined SS segments caused a significant transcriptional change when examined in K562 cells, indicating strong tissue specificity of the tested SSs, limited to GM12878 cells. However, it's not strong evidence to support tissue specificity. As 1) the data the author used to predict SSs is only limited to GM12878 cells and there's no data shown how H3K27me3 signals are in K562 cells 2) The experiment is only done in both GM12878 and K562, there's no other cells tested. It's not a reasonable assumption.

The author investigated the molecular mechanism that converts GM12878 SSs to cancer SE regions. It would be interesting to see how the conversion from SE to SSs occurs. A comparison study is highly recommended.

There are a lot of studies about super-enhancers, and as super-silencer is a converse concept, the author could check if SSs interact or not on the scale of chromatin 3D structure.

Minor Comments:

In the result section, paragraph 1, the author mentioned nearly 66% of constituent silencers in our SS regions overlapped with MRRs reported in the previous study. But there's no content to explain what MRRs are. In Figure 7B, it is not clear what the y-axis represents.

Version 1:

Reviewer comments:

Reviewer #1

(Remarks to the Author)

My comments have been addressed and I think the manuscript should be accepted. I congratulate the authors on an interesting study!

Reviewer #2

(Remarks to the Author)

My concerns have been satisfactorily addressed.

Reviewer #4

(Remarks to the Author)

The authors have largely responded well to the criticism by the reviewers. A remaining weak point is the luciferase assay to analyze silencing properties of SS elements. Even if this assay has been used by many others, it remains inappropriate in this context, as it fails to record almost any of the in vivo parameters, which might influence the outcome. One potential way out of this dilemma could be to use episomal vectors to ensure that the silencing assay includes chromatin structures established in cycling cells.

Responses to reviewers

We express our gratitude to the reviewers of our manuscript for their invaluable feedback and suggestions, which were essential in guiding our revisions.

In the revised manuscript, we included experimental validation results on the super-silencers within the BCL6 locus in both GM12878 and primary B cells.

For the analysis, we utilized the DLBCL SEs identified in a recent study by Bal et al. (Bal et al. 2022), which reported SEs from 29 DLBCL cell lines. These cell lines, including BJAB, HBL1, Karpas-422, Ly1, Ly3, Ly7, NU-DHL1, Pferffer, SU-DHL2, SU-DHL-6, WSU-DLCL2, etc., represent the major DLBCL cell-or-origin subtypes – 20 germinal-center-B-cell-like DLBCL and 9 activated-B-cell-like DLBCL cell lines. The analysis results based on these SEs consistently align with the original ones [which were based on the SEs reported in the Chapuy's study (Chapuy et al. 2013)].

We expanded the analyses on the conversions of SSs to lymphoma SEs. The newly-added results illustrate that, among enhancers, silencers and CTCF binding sites, the genomic regions undergoing these conversions: 1) are associated with the most significant upregulation of gene expression in lymphoma; 2) exhibit the most significant hypermutation levels; and 3) show the most significant alteration in chromatin contacts. These findings suggest the pivotal roles of these conversions in lymphoma carcinogenesis.

Additionally, we have extended the analyses to other cell lines, including hESC H1, HepG2 and K562. In these cell lines, the predicted SSs are consistently proximal or linked to the genes with the lowest expression levels and are enriched with the repressive elements identified by massively parallel reporter assays.

Moreover, we analyzed SE-to-SSs, which are conversions opposite to SS-to-SEs, in B cell lymphoma cell lines. Based on the H3K27me3 data availability in the ENCODE project, we identified the SSs in six B-cell cancer cell lines, including Ly1 (ENCODE biosample EFO:0005907), Ly3 (EFO:0006710), Karpas-422 (EFO:0005719), SU-DHL-6 (EFO:0002357), etc., outlining the similarities and differences between SE-to-SSs and SS-to-SEs in the revised manuscript.

Our point-by-point responses below provide a summary of the changes made.

REVIEWER COMMENTS

Reviewer #1 (Remarks to the Author):

In this manuscript, Huang et al investigate putative supersilencers and their roles and potential mechanisms in GM12878 and B cell lymphoma cell line data. While the analyses shown here largely make sense, and there is great interest in the growing field of supersilencers, there is very little wet lab experimental testing, meaning that many conclusions are not rigorously supported.

We added the experimental data in GM12878 and primary B cells obtained from healthy human donors enrolled under NIH Clinical Center Institutional Review Board-approved protocol 99-GC-0168.

Additionally, a lot is over-claimed on the basis of GM12878 and a few cell lines, but usually to make claims about cancer, more clinical samples (both cancer and normal) are needed.

We added the analysis results for primary B cells (ENCODE ID ENCBS857XIR), demonstrating the significant similarity in silencer and enhancer profiles between GM12878 and primary B cells. Additionally, we used DLBCL SEs identified in a recent study by Bal et al. (Bal et al. 2022). DLBCL samples used in this study represent the major DLBCL cell-or-origin subtypes. The analysis results based on these SEs consistently align with those presented in the original version of the manuscript [which were based on the SEs reported in the Chapuy's study (Chapuy et al. 2013), as detailed below.

Furthermore, we have toned down our statements. For example, we have changed the title of this manuscript into "Super-silencers are crucial for development and carcinogenesis in B cells." and added the following lines (Page 23) to outline limits of this study.

As demonstrated in this study for primary B cells and supported by previous research, the GM12878 cell line effectively recapitulates the cellular behaviors and molecular characteristics of primary B cells³⁹⁻⁴². Its vast sequencing data resources and ease of targeted modifications further establish its widespread use as a surrogate for primary B cells. However, experimental validation in primary B cells and in vivo conditions may be necessary to ensure the broader applicability of our findings to different B cell subtypes across varied cellular contexts

Other major comments

1. The two models (Figure 8) are interesting but there is insufficient experimental evidence provided for them. Ideally multiple lines of experimental evidence should support these two models.

We appreciate the reviewer's comment and have examined two main repression models in three additional cell types K562, HepG2, hESC H1. These cell types were chosen based solely on the availability of histone ChIP-seq data (for building silencer prediction models) and chromatin contact data (for testing repression models). As shown in Figure R1, in K562, HepG2 and hESC H1 cell lines,

1. CGI-SSs and their adjacent elements (within ± 50 kbp) show a higher Hi-C contact density than non-CGI counterparts (the plots in the left column).
2. Target promoters of CGI SSs show high contact abundance (the plots in the middle column), but the depletion of enhancer contacts (the plots on the right).

These results are consistent with the observations in GM12878.

Figure R1 Repression models of nonCGI and CGI SSs in three cell types (A) HepG2, (B) K562, and (C) hESC H1. The dashed lines are the expected values across the whole genome.

We have included this figure along with the following statement on Page 20.

“These trends were also observed in other cell types, including H1 hESC, HepG2 and K562. Specifically, CGI SSs and their adjacent regions have more Hi-C contacts than non-CGI SSs ($P < 10^{-10}$ in most of the comparisons, the plots in the left column in Fig. S21). On the other hand, the promoters linked with CGI SSs show increased densities of total contacts but decreased densities of enhancer contacts ($P < 10^{-10}$, the plots in the middle and right columns, Fig. S21).”

2. Only GM12878 is used as a “normal” B cell. It is not even very normal as it grows in culture and is EBV infected. More experimental data from normal B cells should be given.

We valued the reviewer’s comment and experimentally examined the activity of six segments of the BCL6 SS in both GM12878 and primary B cells. Three of these segments function as silencers in primary B cells (Figure R2A), of which two are silencers in both GM12878 and primary B cells. Furthermore, to assess the regulatory similarity between GM12878 and primary B cells, we compared silencer and enhancer profiles between GM12878 and primary B cells (ENCODE biosample ID ENCBS857XIR). After identifying super silencers and typical silencers (and super enhancers and typical enhancers) in primary B cells, we observed that 36.7% of GM12878 super silencers acting as super silencers in primary B cells. This fraction significantly exceeds the average 6.7% observed in HepG2, H1, and K562 (Figure R2B). Similarly, GM12878 TSs, TEs and SSs are significantly enriched among their counterparts in primary B cells. These significant enrichments evident GM12878 as a suitable model for studying B cell gene regulation. GM12878 has been widely considered as the common choice for B cell analysis (Zhao et al. 2014; Jiang et al. 2023), frequently used as a reference point for normal B cell behavior in B cell cancer studies (Almamun et al. 2014; Li et al. 2021).

Figure R2 Experimental results of SS segments in the *BCL6* locus in (A) GM12878 and primary B cells. (B) Fractions of GM12878 silencers and enhancers displaying the same activity in other cell types. “BK” represents the cell types hESC H1, HepG2 and K562, while “B cell” represent the primary B cells.

We have included these figures along with the following statement on Page 9.

Additionally, among the six examined SS segments located in the BCL6 locus, three exhibit repressive regulatory activity in GM12878 cells ($P \leq 0.05$ vs control elements, Fig. S9). Combined, 70% of ten examined SS segments in the BACH2 or BCL6 locus were experimentally validated as silencers in GM12878. Moreover, three out of six GM12878 SS segments function as a silencer in primary B cells (Fig. S9A). Two of them (SS8 and SS9) have repressive influence in both GM12878 and primary B cells, hinting a high degree of similarity in silencer profiles between these two B cell types. To further assess this similarity, we compared the silencer profiles between GM12878 and the primary B cell biosample documented in the ENCODE project (see Supplementary Notes). On average, over 33% of GM12878 SSs function as SSs in primary B cells, a significantly greater overlap compared to only 6.7% of these elements acting as SSs in other cell types, including hESC H1, HepG2, and K562 ($P = 0.004$, Fig. S9B). Similarly, TSs, SEs and TE profiles in the GM12878 display substantial overlaps with their counterparts in primary B cells ($P \leq 0.04$). Notably, GM12878 SSs and SEs, the regulatory element types critical for maintaining cell identity, show more enrichment among their counterparts in primary B cells than TSs and TEs. These high similarities underscore the GM12878 cell line as a suitable model for studying gene regulation in B cells^{39,40} and as a frequently used reference point for normal B cell behavior in B-cell cancer studies^{41,42}.

3. More B cell lymphoma results should be included. Also, are there controls for batch effects because these different datasets may have come from different sources?

We have analyzed 10 DLBCL samples (6 cell lines and 4 patients) and 18 CLL patients in our original study. During the revision, we included the analysis based on the super-enhancers reported in the Bal’s study (Bal et al. 2022). This study reported the super-enhancers in 29 DLBCL cell lines, including BJAB, HBL1, Karpas-422, Ly1, Ly3, Ly7, NU-DHL1, Pferffer, SU-DHL2, SU-DHL-6, WSU-DLCL2, etc. The analysis results on these super-enhancers are consistent with the original ones.

We added the following lines (Page 15).

“Consistently, 670 B cell SS components (14.5%) overlap with DLBCL SEs reported by Elodie Bal et al.⁶⁰, a fraction significantly higher than 7.9% of GM12878 TSs and 3.2% of the background sequences (binomial test $P < 10^{-10}$, Fig. S14C).”

4. Clinical B cell lymphoma results should also be given. The data downloaded from GEO comes from just a few cell lines, as far as I am aware.

- DLBCL SEs explored here were detected in two studies: the Chapuy et al.'s study and the Bal et al.'s study. The study of Chapuy et al. 2013 involved 4 patients and 6 DLBCL cell lines, as listed below.

Patient	study
GCB#1	Chapuy et al. 2013
GCB#2	Chapuy et al. 2013
ABC#1	Chapuy et al. 2013
ABC#2	Chapuy et al. 2013
Cell line	
Ly1	Chapuy et al. 2013
DHL6	Chapuy et al. 2013
Ly3	Chapuy et al. 2013
HBL1	Chapuy et al. 2013
Ly4	Chapuy et al. 2013
Toledo	Chapuy et al. 2013

The study of Bal et al. 2022 identified SEs from 29 DLBCL cells, including BJAB, DB, DOHH2, FARAGE, HBL1, HLY1, HT, Karpas-422, Ly1, Ly3, Ly7, Ly8, Ly10, Ly18, NU-DHL1, Pfeiffer, RCK8, RIVA, RL, SU-DHL2, SU-DHL5, SU-DHL6, SU-DHL7, SU-DHL10, TMD8, Toledo, U2832 and WSU-DLCL2. The samples represent the major DLBCL cell-or-origin subtypes – germinal-center-B-cell-like DLBCL and activated-B-cell-like DLBCL cell lines.

- CLL SEs were detected from 18 “primary B-cell samples purified from the peripheral blood of treatment-naïve individuals diagnosed with CLL”, as mentioned in the CLL paper (Ott et al. 2018).

Patient	study
CLL#1	Ott et al. 2018
CLL#2	Ott et al. 2018
CLL#3	Ott et al. 2018
CLL#4	Ott et al. 2018
CLL#5	Ott et al. 2018
CLL#6	Ott et al. 2018
CLL#7	Ott et al. 2018
CLL#8	Ott et al. 2018
CLL#9	Ott et al. 2018
CLL#10	Ott et al. 2018
CLL#11	Ott et al. 2018
CLL#12	Ott et al. 2018
CLL#13	Ott et al. 2018
CLL#14	Ott et al. 2018
CLL#15	Ott et al. 2018
CLL#16	Ott et al. 2018
CLL#17	Ott et al. 2018
CLL#18	Ott et al. 2018

- We have added the descriptions of these samples in Table S4 and described these samples in the section Methods as follows (Page 26).
“DLBCL SEs were detected in two independent studies, one by Chapuy et al.³⁸ and the other by Bal et al.⁶⁴. The former involved 4 DLBCL patients and 6 DLBCL cell lines, while the latter investigated 29 DLBCL cell lines. CLL SEs were detected from 18 patients by Ott et al.⁶⁹. Information of these samples is summarized in Table S4.”

5. The title is too broad considering the results shown - not all of carcinogenesis is shown. Only B-cell lymphoma is discussed.

We valued the reviewer’s comment and have changed the title to “Super-silencers are crucial for development and carcinogenesis in B cells.”, emphasizing our findings in B cells and B cell cancers.

6. Luciferase reporter is an in vitro assay and somewhat artificial. More could be done (eg CRISPR excision or modulation) for more biologically relevant results.

Luciferase-based assays are widely used to experimentally evaluate the activity of enhancers and silencers. For example, in the Pang’s study (<https://www.nature.com/articles/s41588-020-0578-5>), luciferase assays were used to verify the silencers identified in the ReSE platform in K562 and HEK 293T cell lines, as shown below.

Also in the Zhu’s study (<https://www.biorxiv.org/content/10.1101/2023.06.20.545673v1.full.pdf>), luciferase assays were used to verify the silencers identified in a MPRA platform in K562, LNCap, and HEK 293T cell lines, as shown below.

Figure 6: Luciferase assays to determine the repressive activity of silencers identified from HEK 293T, LNCap and K562 cells. The empty PGL 4.53 plasmid was the control group. The y axis represents the percentage of luciferase activity compared to pGL4.53 empty plasmids in the respective cells (3 biological replicates per sample; bars show mean value±s.e.m. *: p<0.05, **:p<0.01, ns: no significant difference, calculated using t-test).

Here, we conducted in vitro validations to GM12878 and primary B cells, confirming the effectiveness of luciferase assays in these cell types.

Reviewer #2 (Remarks to the Author):

Super silencer (SS) is an emerging concept that describes genomic regions with the most density in histone repressive marker (H3K27me3). The authors performed a comprehensive analysis on SS in B cell lymphoma using public data generated for GM12878. The analysis showed that SSs are strongly associated with developmental-specific genes, enriched with cancer-specific SNPs/variants, and linked to conversion to cancer-specific super enhancers. The authors further found that SS are enriched at TAD shores unique to B-cell cancers, supporting a potential role in sharpening the 3D chromatin architecture in cancer cells. Generally, the experiments are well designed, the data analysis is well controlled, and the results are supported by the analysis. I have two points for the authors to consider.

1) A clear distinguishing of the current manuscript from two previous manuscripts (refs 24 and 25) should be fully discussed in the discussion section.

We value the reviewer's comment and have expanded the discussion on these related studies. We have also emphasized our novel findings, which can be summarized as 1) SSs are crucial for development and maintenance of cell identity; 2) SSs play an essential role in maintaining chromatin structure; 3) conversion of SSs to SEs is a key factor in DLBCL carcinogenesis.

We have included the following paragraphs in Discussion.

“Silencers have been observed to form genomic regions that are characterized by exceptionally high H3K27me3 signals and notable repressive influence in gene regulation^{24,25}. In the K562 cell line, the knockout of the active region of a SS (or referring to as an H3K27me3-rich region) results in the upregulation of the target genes, including FGF18, a gene involved in cell differentiation and cell migration, and subsequently inhibits tumor growth^{24,25}. In this study, we have identified SSs in difference cell lines as the regions having the densest concentrations of silencers and carrying significant H3K27me3 signals. We further showed that, in differentiated B cells, SSs downregulate genes highly expressed during development, particularly in early developmental stages. These SSs are notably enriched in gene loci essential to B cell differentiation and development, including BCL6 and FOXO1, while being absent in the tumor repressor genes, such as TP53 and NCOR1. This suggests that SSs may be essential in defining cell identity by suppressing development-specific genes.

To elucidate the role of SSs in the maintenance of cell identity, we explored their association with chromatin architecture, considering that 1) 3-D chromatin folding is crucial for establishing and maintaining cell identity^{82,83}; and 2) H3K27me3 serves as a mediator of chromatin architecture⁸⁴. In the K562 cell type, deactivating a SS results in the loss of the local TAD and chromatin loops²⁴. Our study demonstrates the abundance of SS components in proximity to TAD boundaries. These TAD-boundary-proximal SS components are uniquely enriched in B-cell cancer TLBPs, second only to TAD-boundary binding sites of CTCF, the most well-known 3D-genome regulator. In addition, during the carcinogenesis of B cell lymphoma, SS components exhibit more loss in chromatin contacts than TSs, enhancers, and CTCF binding sites. Collectively, our findings strongly support the pivotal role of SSs in shaping and maintaining the genome's TAD architecture.

Emerging evidence indicates that cancer progression is associated with the activation or acquisition of SE^{85,86}. Although rarely explored, if at all, the conversions from SSs to cancer SEs occur on a large scale. Importantly, the genomic regions undergoing these conversions are associated with remarkable upregulation of gene transcription. They exhibit the strikingly high density of somatic variants of B cell cancers. Furthermore, these regions display the most significant loss in chromatin contacts in naïve DLBCL cells and the most significant re-establishment in chromatin structure after deactivating enhancers in DLBCL cells, among examined regulatory elements, including CTCF binding sites. Our findings repeatedly underscore the crucial contribution of the conversions from SSs to cancer SEs to carcinogenesis, partly through reorganizing the 3-D chromatin structure, a well-known underpinning of carcinogenesis⁸³. ”

2) The analysis on transcription response to JQ1 for SS-associated genes could be extended to the context of 3D chromatin. It is not clear whether the “SS-to-SE” genes include those identified through 3D chromatin-chromatin interaction or not. If not, that will be something to enhance the conclusions of the manuscript for the role of SS in modulating TAD structures.

We agreed that analyzing chromatin structure would be beneficial for our study. We conducted a comparison of chromatin contacts among GM12878, naïve and A-485-treat Karpar-422 cells. Similar to JQ1, A-485 deactivates or weakens enhancers. Therefore, these comparisons can offer insights into the contribution of SSs, especially SS-to-SEs, to the organization of chromatin architecture. As shown in Figure R2, GM12878 SSs lost more chromatin contacts in Kapar-422 cells than TSs, CTCF binding sites and enhancers. These suggest that SSs, particularly SS-to-SEs, are pivotal in maintaining chromatin structure.

Figure R3 Fractions of significant chromatin contacts conserved between GM12878 and (A) untreated or (B) A-485-treated Karpas-422 cells. ** $p < 10^{-10}$.

We added this figure and the following discussions in the manuscript (Page 15).

“To investigate the impact of SS-to-SE conversions, we examined the changes in chromatin contacts linked to these regions during carcinogenesis. Approximately 9% of contacts to SS components in GM12878 cells persist in naïve Karpas-422 cells, a fraction significantly lower than the expected 12% by chance and 14% of contacts to enhancers or CTCF binding sites ($P = 0.001$, Fig. S15A). Notably, this fraction diminishes further to 5% among the contacts to SS-to-SEs. Conversely, 35% of SS-to-SE GM12878 contacts are retained in the Karpas-422 cells treated with A-485, a small molecule deactivating enhancers by inhibiting the activity of histone acetyltransferases p300 and CBP⁶⁵. This fraction significantly exceeds the 31% of contacts to CTCF binding sites and 18% of contacts to DLBCL SEs ($P < 0.001$, Fig. S15B). These findings suggest that SS-to-SE conversions frequently reorganize chromatin folding, a major factor contributing to gene deregulation during carcinogenesis.”

Reviewer #3 (Remarks to the Author):

The study explores variations in the repressive H3K27me3 signals in silencers, emphasizing 'super-silencers.' These regions were initially linked to gene upregulation during development and impact B-cell function. Some super-silencers transform into super-enhancers in B-cell lymphoma, with key genes responding more rapidly to cancer therapy. Super-silencers also exhibit an overabundance of B-cell cancer-related mutations and translocation breakpoints, underscoring their pivotal role in B-cell regulation. The CpG content in super-silencers influences their repressive mechanisms, with implications for B-cell carcinogenesis.

Major Comments:

The author predicted SSs and SE in the GM12878 cell line and shows low DNA methylation levels, CTCF binding, and TADs among all those regions. It would be interesting to check gene deserts and housekeeping genes among those regions.

We appreciated the Reviewer's comments and have examined the distribution of SSs in gene deserts and the loci of housekeeping genes. Utilizing the gene annotations from the GENCODE version 45 (https://www.genecodegenes.org/human/release_45lift37.html), we extracted the 662 gene desert regions in the human genome, defined as genomic regions over 500mb in length containing no protein-coding genes. We noticed that both TSs and TEs are more frequently located within these gene deserts than their super element counterparts in all examined cell lines (Figure R4A). We identified 1,790 housekeeping and 2,064 cell-specific genes, based on the gene expression profiles across 251 biosamples in the ENCODE project. Our analysis revealed that SS components exhibit the lowest enrichment level in housekeeping gene loci but the highest enrichment in cell-specific gene loci in all tested cell lines (Figures R4B and C).

Figure R4 Distribution preference of SSs across cell lines. Fraction of elements in (A) gene deserts; (B) cell-specific gene loci; (C) housekeeping gene loci. ** $p < 10^{-10}$.

We have included these results and relevant discussions in the manuscript as follows (Page 7).
“We further examined the predicted SS components from additional views. Firstly, TSs and TEs show a higher density in gene deserts than their super element counterparts, especially in the stem cell line hESC H1 (Fig. S7A). Secondly, 20.8% of SS components are proximal to cell-specific genes, significantly exceeding 14.7% expected among TSs and 11.8% among enhancers (binomial test $P < 10^{-10}$, Fig. S7B). In contrast, SS components are notably less abundant in housekeeping gene loci (6.4% of SS components vs 9.5% of TSs vs 15% of enhancers, $p < 10^{-10}$, Fig. S7C).”

The author mentioned testing silencing activity using luciferase reporter assays. And none of the examined SS segments caused a significant transcriptional change when examined in K562 cells, indicating strong tissue specificity of the tested SSs, limited to GM12878 cells. However, it's not strong evidence to support tissue specificity. As 1) the data the author used to predict SSs is only limited to GM12878 cells and there's no data shown how H3K27me3 signals are in K562 cells 2) The experiment is only done in both GM12878 and K562, there's no other cells tested. It's not a reasonable assumption.

We agree with the reviewer's comment that experimental luciferase results of four tested SS fragments in two cell types (i.e., GM12878 and K562) are not sufficient to support high cell-specificity of SSs. Thus, we have removed the statement of "indicating the tissue-specificity of the tested SSs."

Also, we predicted the SSs in three additional cell types (i.e., hESC H1, HepG2, and K562), and validated the negative regulatory impact of the predicted SSs based on 1) the expression of their proximal or target genes; 2) experimental results from MPRAs, as shown in Figure R5.

Figure R5. Validation of the predicted SSs in three cell lines, hESC H1, HepG2 and K562 using (A) their proximal genes; (B) their Hi-C contact genes; (C) the results from Sharpr MPRAs. ** $p < 10^{-10}$.

Comparing the SS profiles in GM12878 and three newly added cell types, we noticed the slightest overlap among SSs in these cell types (Figure R6). Despite the limited examination, our observations hint the high cell-specificity of silencers, especially SSs.

Figure R6. Overlap among silencers and enhancers (A) across four examined cell lines, i.e., GM12878, H1, HepG2 and K562; (B) between GM12878 and B cell cancer cell lines. The overlap was measured in term of the Jaccard index.

We have included these results as follows (Page 7).

"We expanded our analysis to include three additional cell types (i.e., hESC H1, HepG2, and K562). On average, we identified about 6,000 SS components per cell type. In these cell types, genes proximal to or linked with these elements consistently exhibit the most significant downregulation

levels among all examined genes ($P < 10^{-10}$, Fig. S6). Furthermore, in the cell types K562 and HepG2 where the regulatory impact of genome sequences has been assessed in Sharpr MPRA^s ³⁵, the predicted SS components and TSs are significantly enriched with negative Sharpr scores, while predicted enhancers exhibit positive Sharpr scores on average ($P < 10^{-10}$, SS components and TSs vs SE components and TEs, Fig. S6). These scores support the repressive impact of silencers and activating influence of enhancers. These results, aligning with those observed in GM12878 (Fig. 1), support the negative regulatory function of predicted silencers, especially SSs, across different cell types.

... ..

the Jaccard indices of SS components across these cell types (i.e., the fractions of SS components shared between in these cell types) average 0.039, significantly lower than 0.095 of TSs, 0.084 of SEs, and 0.129 of TEs (binomial test $p \leq 0.01$, Fig. S8A). Additionally, the Jaccard indices between GM12878 SSs and B cell cancer cell lines average 0.034, significantly lower than those for enhancers (binomial test $p \leq 0.01$, Fig. S8B). These results consistently hint the high specificity of silencers (together with their associated genes), particularly SSs, to one examined cell type or cellular context.”

The author investigated the molecular mechanism that converts GM12878 SSs to cancer SE regions. It would be interesting to see how the conversion from SE to SSs occurs. A comparison study is highly recommended.

We valued the reviewer’s suggestion. After profiling SSs in B cell cancer cells (including Ly1, Ly3, Karpas-422, SU-DHL-6, HL-60, and MM.S1), we identified 484 SE-to-SSs, i.e., GM12878 SE components acting as SSs in B cell cancer cells. Similar to the conversion from SSs to cancer SEs, the SE-to-SS conversion occurs more frequently than the expected values among TEs or by chance (Figure R7).

Figure R7. Function of GM12878 silencers and enhancers in B cell cancer cells. (A) Fraction of these elements acting as SEs in B cell cancer cells. (B) Recurrent rate of the elements acting as SEs. ** $p < 10^{-10}$; * $p \leq 0.01$.

Also, like SS-to-SEs, SE-to-SSs frequently lose chromatin contacts during DLBCL development (Figure R7).

Figure R8. Fraction of GM12878 chromatin contacts conserved in Karpar-422, a DLBCL cell line.

We have included these results as follows (Page 17).

“In addition, we investigated the conversion of SEs to SSs, which represents the extreme opposite to the SS-to-SE conversion, during B cell carcinogenesis (Supplementary Notes). Of GM12878 SE components, 4.8% function as a SS in at least one examined B-cell cancer cell type, significantly exceeding 4.3% observed in the background regions or TSs (binomial test $P \leq 0.01$, Fig. S18A). Furthermore, 7.4% of these SE-to-SSs recur, exceeding the 5.0% recurrence rate among the background sequences ($P < 10^{-10}$, Fig. S18B). Furthermore, SE-to-SSs exhibit greater loss in chromatin contacts than the other SEs ($P = 0.0001$, Fig. S15A). These findings, along with the high recurrence of SS-to-SE conversions (Fig. 6), suggest that 1) large-scale functional overturns occur during DLBCL development; and 2) these overturns often coincide chromatin reorganizations. However, unlike SS-to-SEs which typically harbor an exceptional density of cancer SNVs, SE-to-SSs exhibit a density of B-cell lymphoma SNVs similar to other GM12878 SEs (Fig. S14B). These results suggest that SE-to-SS conversions are less likely to play a dominant role in B cell lymphoma.”

There are a lot of studies about super-enhancers, and as super-silencer is a converse concept, the author could check if SSs interact or not on the scale of chromatin 3D structure.

This is an important topic in SS analysis. While we originally demonstrated that SSs and SEs tend to interact with the elements of same types (Fig. 8A): SSs interact with SSs more frequently than with other elements, and SEs more frequently with SEs. We also showed that the promoters linked to SSs are depleted of enhancer (including SEs) contacts (Fig. 8D).

However, it is noteworthy that 12 gene promoters are linked to both SSs and SEs in GM12878. The Gene Ontology analysis showed that these genes, including RARA, NEBAL2, ZNF358, ATG5, are associated with immune system development and hematopoietic organ development. We have included these findings in the manuscript as follows (Page 20).

“Moreover, the gene promoters targeted by SSs show a reduction in enhancers (including SEs) contacts (Fig. 8D). Despite their scarcity, it is worth mentioning that 12 gene promoters are linked to both SSs and SEs. Among these genes are RARA, NEBAL2, ZNF358, and ATG5, all of which contribute to immune system development and hematopoietic organ development⁸⁰.”

Minor Comments:

In the result section, paragraph 1, the author mentioned nearly 66% of constituent silencers in our SS regions overlapped with MRRs reported in the previous study. But there's no content to explain what MRRs are.

MRRs, i.e., H3K27me3-rich regions, are the genomic regions exhibiting the significant intensity of H3K27me3 modification, as reported in Cai's study. They are essentially SSs by a different name. We thereby compared our SSs with MRRs.

We have included the following for clarification.

"Notably, nearly 66% of constituent silencers in our SS regions overlapped with H3K27me3-rich regions (MRRs) reported in the previous study"²⁵, ..."

In Figure 7B, it is not clear what the y-axis represents.

The heatmap in Figure 7B represents the enrichment levels of silencers and enhancers in TAD boundaries, and the regions proximal to TAD boundaries (i.e., 20kbs to 50kbs from a TAD-boundary, denoted as TAD shores in this study). Each row represents a TAD boundary.

We added the following line in the caption of Figure 7B for further explanation.

"Each row in this heatmap represent a TAD boundary. Enrichment fold is calculated through comparing with the whole genome."

- Almamun M, Levinson BT, Gater ST, Schnabel RD, Arthur GL, Davis JW, Taylor KH. 2014. Genome-wide DNA methylation analysis in precursor B-cells. *Epigenetics* **9**: 1588-1595.
- Bal E, Kumar R, Hadigol M, Holmes AB, Hilton LK, Loh JW, Dreval K, Wong JCH, Vlasevska S, Corinaldesi C et al. 2022. Super-enhancer hypermutation alters oncogene expression in B cell lymphoma. *Nature* **607**: 808-815.
- Chapuy B, McKeown Michael R, Lin Charles Y, Monti S, Roemer Margaretha GM, Qi J, Rahl Peter B, Sun Heather H, Yeda Kelly T, Doench John G et al. 2013. Discovery and Characterization of Super-Enhancer-Associated Dependencies in Diffuse Large B Cell Lymphoma. *Cancer Cell* **24**: 777-790.
- Jiang K, Fu Y, Kelly JA, Gaffney PM, Holmes LC, Jarvis JN. 2023. Comparison of the three-dimensional chromatin structures of adolescent and adult peripheral blood B cells: implications for the study of pediatric autoimmune diseases. *bioRxiv* doi:10.1101/2023.09.11.557171.
- Li X, Duan Y, Hao Y. 2021. Identification of super enhancer-associated key genes for prognosis of germinal center B-cell type diffuse large B-cell lymphoma by integrated analysis. *BMC Medical Genomics* **14**: 69.
- Ott CJ, Federation AJ, Schwartz LS, Kasar S, Klitgaard JL, Lenci R, Li Q, Lawlor M, Fernandes SM, Souza A et al. 2018. Enhancer Architecture and Essential Core Regulatory Circuitry of Chronic Lymphocytic Leukemia. *Cancer Cell* **34**: 982-995.e987.
- Zhao B, Barrera LA, Ersing I, Willox B, Schmidt SC, Greenfeld H, Zhou H, Mollo SB, Shi TT, Takasaki K et al. 2014. The NF- κ B genomic landscape in lymphoblastoid B cells. *Cell Rep* **8**: 1595-1606.